# Editing of endogenous tubulins reveals varying effects of tubulin posttranslational modifications on axonal growth and regeneration

**Yu-Ming Lu[1], Shan Yan[2], Shih-Chieh Ti[2], Chaogu Zheng[1]\***

[1]School of Biological Sciences, Faculty of Science, The University of Hong Kong, Hong Kong SAR, Hong Kong, China; [2]School of Biomedical Sciences, Li Ka Shing Faculty of Medicine, The University of Hong Kong, Hong Kong, China

**Abstract** Tubulin posttranslational modifications (PTMs) modulate the dynamic properties of microtubules and their interactions with other proteins. However, the effects of tubulin PTMs were often revealed indirectly through the deletion of modifying enzymes or the overexpression of tubulin mutants. In this study, we directly edited the endogenous tubulin loci to install PTM-mimicking or -disabling mutations and studied their effects on microtubule stability, neurite outgrowth, axonal regeneration, cargo transport, and sensory functions in the touch receptor neurons of *Caenorhabditis elegans*. We found that the status of β-tubulin S172 phosphorylation and K252 acetylation strongly affected microtubule dynamics, neurite growth, and regeneration, whereas α-tubulin K40 acetylation had little influence. Polyglutamylation and detyrosination in the tubulin C-terminal tail had more subtle effects on microtubule stability likely by modulating the interaction with kinesin-13. Overall, our study systematically assessed and compared several tubulin PTMs for their impacts on neuronal differentiation and regeneration and established an in vivo platform to test the function of tubulin PTMs in neurons.

**\*For correspondence:**
cgzheng@hku.hk

**Competing interest:** The authors declare that no competing interests exist.

## eLife assessment

This **fundamental** study analyzes the roles of post-translational modifications of tubulin by generating a large panel of tubulin mutants and describing their effects on morphogenesis and function of sensory neurons in *C. elegans*. The work, which is of interest to all cell biologists, in particular researchers with an interest in the microtubule cytoskeleton and neurobiology, presents conclusions that are supported by **solid** evidence. Demonstrating that all introduced mutations have the intended consequences and exploring their direct effect on microtubules would further increase the impact of the work.

## Introduction

Microtubules (MTs) play important roles in neuronal development by providing structural support for neurite growth and serving as tracks for intracellular transport. MTs are formed by the polymerization of α/β-tubulin heterodimers, which are crucial determinants of MT properties. Eukaryotic genomes contain multiple α- and β-tubulin genes (called isotypes) with distinct expression patterns and dynamic properties (*Janke and Magiera, 2020*; *Lu and Zheng, 2022*; *Nsamba and Gupta, 2022*). These tubulin isotypes are also subjected to a range of post-translational modifications (PTMs), which regulate the stability of MTs and their interaction with various microtubule-associated proteins (MAPs;

*Roll-Mecak, 2020*). The findings of multiple tubulin isotypes with different PTMs led to the concept of a 'tubulin code', which suggests that the structure, dynamics, and functions of individual MTs are controlled by the tubulin isotype composition and the tubulin PTMs (*Janke, 2014*). Perturbation to the tubulin code can lead to MT dysfunction and are linked to human diseases (*Tischfield et al., 2011*).

Although previous research described the effects of several tubulin PTMs in neuronal development, these earlier studies have two potential limitations. First, the role of tubulin PTMs was often revealed by the overexpression of tubulin mutants with PTM-mimicking or unmodifiable amino acid substitutions (*Shida et al., 2010*). Such overexpression may create artifacts. Moreover, since the tubulin concentration in cells is controlled by autoregulation through mRNA degradation (*Lin et al., 2020*), overexpression of exogenous tubulins may trigger the downregulation of endogenous tubulin isotypes, complicating the interpretation of the results. Second, the effects of tubulin PTMs were also identified by studying the enzymes that add or remove specific PTMs. In some cases, the phenotypes caused by deleting the enzyme may not be the same as the elimination of the tubulin PTMs, because the enzyme may have multiple substrates, including non-tubulin substrates, or the enzyme has additional functions independent of its activity in modifying tubulins. For example, the function of α-tubulin acetyltransferase MEC-17 in regulating neurite outgrowth is independent of its enzymatic activities (*Topalidou et al., 2012*). Given the above two limitations and recent advances in genome editing, we reason that the role of tubulin PTMs can be directly assessed by engineering the endogenous tubulin genes.

In this study, we installed PTM-mimicking or -inactivating mutations in the endogenous loci of the *Caenorhabditis elegans* tubulin genes that are specific for the touch receptor neurons (TRNs) and examined the effects of these mutations on neuronal development. We and others previously established *C. elegans* TRNs as a model to study the effects of tubulin missense mutations on MT stability and neurite growth at the single-cell level (*Lee et al., 2021*; *Savage et al., 1994*; *Zheng et al., 2017*). TRNs are a set of six mechanosensory neurons that detect gentle body touch and have long sensory neurites along the body wall. These neurites are filled with specialized 15-protofilament MTs (*Chalfie and Thomson, 1982*), presumably made of MEC-12/α-tubulin and MEC-7/β-tubulin, which are the dominant tubulin isotypes in the TRNs and are expressed at much (>100 times) higher levels than other tubulin isotypes (*Taylor et al., 2021*). Mutations in *mec-12* and *mec-7* led to the reduction of MT numbers, loss of the 15-protofilament MT structure, alteration of MT stability, and defects in neurite growth and sensory functions in the TRNs (*Zheng et al., 2017*), suggesting that MEC-12 and MEC-7 control MT structure and function in the TRNs.

Moreover, through the analysis of ~100 missense mutations in *mec-12* and *mec-7*, our previous work established three phenotypic categories for tubulin mutations: loss-of-function (*lf*), which caused mild defects in neurite extension, antimorphic (*anti*) gain-of-function, which led to significant reduction in MT stability and caused strong defects in neurite growth, and neomorphic (*neo*) gain-of-function, which resulted in hyperstable MTs and excessive neurite growth (*Lee et al., 2021*; *Zheng et al., 2017*). These distinct impacts of tubulin mutations on MT properties and neurite growth patterns provide a framework to assess and compare the varying roles of different tubulin PTMs in neuronal differentiation.

Here, by editing the endogenous *mec-12* and *mec-7* loci, we systematically studied the effects of β-tubulin S172 phosphorylation and K252 acetylation, α-tubulin K40 acetylation, and tubulin polyglutamylation and detyrosination on neurite growth and regeneration, MT stability, cargo transport, and neuronal functions. We found that the status of β-tubulin S172 phosphorylation strongly affects MT dynamics and neurite development, whereas α-tubulin K40 acetylation did not appear to affect either neuronal morphology or function in any significant way. Polyglutamylation and detyrosination of the tubulin C-terminal tail had more subtle effects on regulating MT stability and limiting ectopic neurite growth likely by modulating the interaction of microtubules with kinesin-13. Tubulin PTMs can have either negative or positive effects on axonal regeneration depending on the type of modifications. Overall, our studies establish a platform to analyze the effects of tubulin PTMs on neuronal differentiation and regeneration through the editing of neuron type-specific tubulin isotypes.

## Results

### β-tubulin S172 phosphorylation inhibits neurite growth

The first PTM we analyzed was the phosphorylation of β-tubulin at the well-conserved serine 172 (S172) residue (*Figure 1—figure supplement 1A*). In mitotic cells, S172 phosphorylation by cyclin-dependent kinase Cdk1 led to the exclusion of the tubulins from the MTs likely because S172 phosphorylation interfered with both GTP binding to β-tubulin and the longitudinal interdimer interaction (*Fourest-Lieuvin et al., 2006*). In neurons, *Ori-McKenney et al., 2016* found that the Minibrain Kinase (MNB) regulated MT dynamics and dendritic morphogenesis of the *Drosophila* class III da neurons, and *Drosophila* MNB was able to phosphorylate porcine β-tubulin at S172 in vitro. These results suggested that β-tubulin S172 phosphorylation may be a regulatory point for MT stability in neurons. However, the role of S172 in neurite growth has not been tested directly.

We created the phospho-mimicking S172E and the nonphosphorylatable S172A mutations in the endogenous *mec-7* locus through CRISPR/Cas9-mediated gene editing. We found that the *mec-7(S172E)* mutation led to severe defects in axonal growth in TRNs, since all neurites of ALM and PLM neurons, the two major TRN subtypes, were significantly shortened in *mec-7(S172E)* mutant animals. The two anteriorly directed neurites of ALM and PLM were termed ALM-AN and PLM-AN; the posteriorly directed neurite of PLM was termed PLM-PN (*Figure 1A–H*). The phenotypes of S172E mutation were similar to our previously characterized *mec-7(anti)* alleles, which strongly suppressed neurite growth by blocking MT polymerization. In contrast, we observed the growth of an ectopic ALM posterior neurite (ALM-PN) in animals carrying the *mec-7(S172A)* mutation; this phenotype was similar to the *mec-7(neo)* alleles, which induced the growth of the ectopic ALM-PN by increasing MT stability. *mec-7(S172A)* mutants showed mild shortening of ALM-AN and PLM-AN, which were also observed with other *mec-7(neo)* alleles (*Zheng et al., 2017*). Being consistent with the genetic nature of gain-of-function mutants, both S172E and S172A mutations were semidominant since heterozygotes showed similar but less severe phenotypes than the homozygotes (*Figure 1—figure supplement 1B–C*).

Moreover, the *mec-7(S172P)* mutants, which we created to mimic a human TUBB2B missense mutation found in patients with cortical malformations (*Jaglin et al., 2009*), showed a *lf* phenotype with moderate defects in neurite growth (*Figure 1B*). Thus, S172E, S172A, and S172P mutations had three distinct phenotypes matching the three categories we previously defined (*Zheng et al., 2017*). Based on these phenotypic similarities, we reasoned that increased phosphorylation (S172E) likely prevented polymerization, leading to severe defects in neurite extension, whereas abolishing phosphorylation (S172A) led to hyperstable MTs and excessive neurite growth. The S172P mutation probably disrupted protein folding or blocked GTP binding, which rendered MEC-7 inactive and caused a *lf* phenotype identical to the *mec-7(-)* mutants (*Zheng et al., 2017*).

To confirm that MEC-7 S172 phosphorylation affects neurite growth cell-autonomously, we extracted the TRNs from the embryos and differentiated them in vitro. Wild-type TRNs grew long neurites in culture, whereas S172E but not S172A mutation significantly shortened the in vitro developed neurites, supporting that S172 phosphorylation controls neuronal morphology in a cell-intrinsic manner (*Figure 1I–J*). We also stained these in vitro cultured TRNs with anti-phospho-tubulin (S172) antibodies and found a considerable reduction of the staining signal in S172A mutant TRNs (*Figure 1—figure supplement 2*). We assumed that the residual signal came from other phosphorylated β-tubulin isotypes expressed in the cells or nonspecific binding of the antibody.

### β-tubulin S172 phosphorylation reduces MT stability

Next, we examined MT dynamics by tracking the plus-end binding protein EBP-2. We found that the wild-type TRNs showed only a few EBP-2 comets due to very stable MTs as previously reported (*Lee et al., 2021*), whereas S172E mutants had increased number of EBP-2 tracks, suggesting increased MT dynamics (*Figure 2A–B*). Moreover, anterior neurites in wild-type TRNs (e.g. ALM-AN) had uniformly plus-end-out MTs, whereas S172E mutants showed mixed MT polarity (*Figure 2C*). The altered MT dynamic properties were consistent with the defects in neurite growth. To assess the difference in MT stability between the wild-type and S172A mutants (both of which showed limited dynamics under normal conditions), we treated the animals with low concentration of colchicine to depolymerize the MTs and then monitored MT dynamics during the recovery phase. Under this sensitized condition, we found that S172A mutants showed fewer EBP-2 comets than the wild-type animals, which suggested reduced MT dynamics and elevated stability (*Figure 2A–B*). Nevertheless, MT polarity was preserved

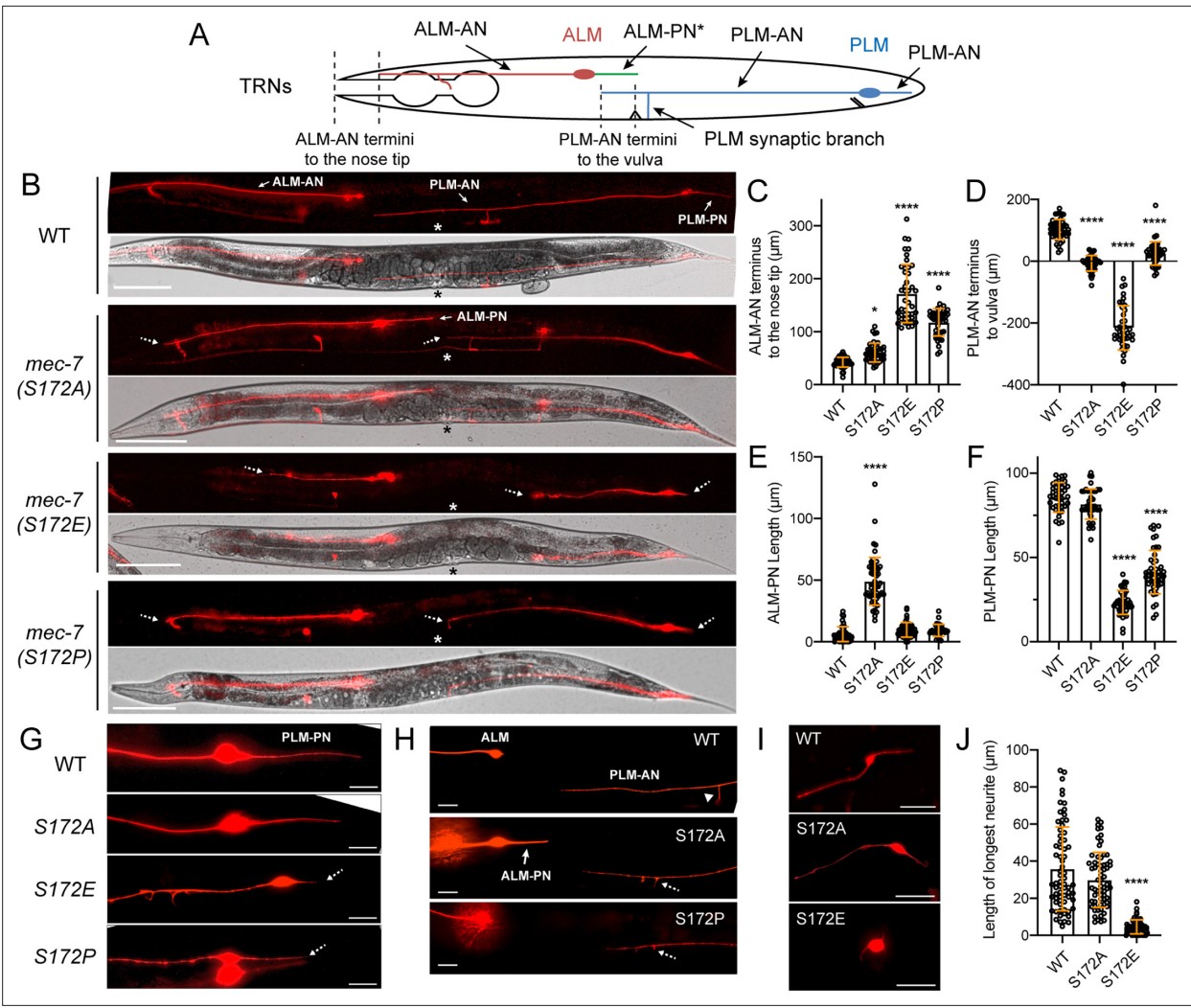

**Figure 1.** Substitution of MEC-7/β-tubulin S172 led to neurite growth defects in *C. elegans* TRNs. (**A**) Neurite morphologies of ALM and PLM neurons. ALM-PN (green) is not prominent in the wild-type animals. ALM-AN and PLM-AN length are measured by the distance of their termini to the nose tip and the vulva (indicated by the distance between the dash lines). (**B**) TRN morphologies in wild-type animals, *S172A*, *S172E*, and *S172P* mutants. Specific neurites are indicated by arrows. Asterisks mark the position of the vulva. Dashed arrows point to the termini of neurites that are shortened in mutants. Scale bar = 100 µm. (**C**) The distance of ALM-AN terminus to the tip of the nose in various strains. The longer the distance, the shorter the ALM-AN. One and four asterisks indicate p<0.05 and 0.0001, respectively, in statistical significance when compared with the wild type in a post-ANOVA Dunnett's test. (**D**) The distance from the PLM-AN terminus to the vulva in various strains. If the anteriorly directed PLM-AN grew past the vulva, the distance is positive. If PLM-AN cannot reach the vulva, the distance is negative. (**E**) Quantification of ALM-PN length. The wild-type animals have no or very short ALM-PN. (**F**) Quantification of PLM-PN length. (**G**) Representative images of PLM-PN in various strains. Dashed arrows indicate the neurite termini of shortened PLM-PN. (**H**) Representative images of ALM-PN in *mec-7(S172A)* mutants. Arrowhead indicates the synaptic branch of PLM-AN in the wild-type animals. Dashed arrows point to the branching defects where the synaptic branch failed to extend to the ventral cord. (**I**) TRNs extracted from the embryos of S172 mutants and cultured in vitro; they were identified by their expression of *mec-17p::TagRFP* among the embryonic cells. TRNs from *S172E* mutants had no or very short neurites. (**J**) The length of the longest neurite of the in vitro cultured TRNs from the S172 mutants. Scale bar = 20 µm.

The online version of this article includes the following source data and figure supplement(s) for figure 1:

**Source data 1.** Numeric data for *Figure 1B–E , and I*.

**Figure supplement 1.** *mec-7* S172A and S172E mutations are semidominant.

**Figure supplement 1—source data 1.** Numeric data for *Figure 1—figure supplement 1B and C*.

**Figure supplement 2.** *mec-7* S172 mutations altered β-tubulin S172 phosphorylation level.

**Figure supplement 2—source data 1.** Numeric data for *Figure 1—figure supplement 2B*.

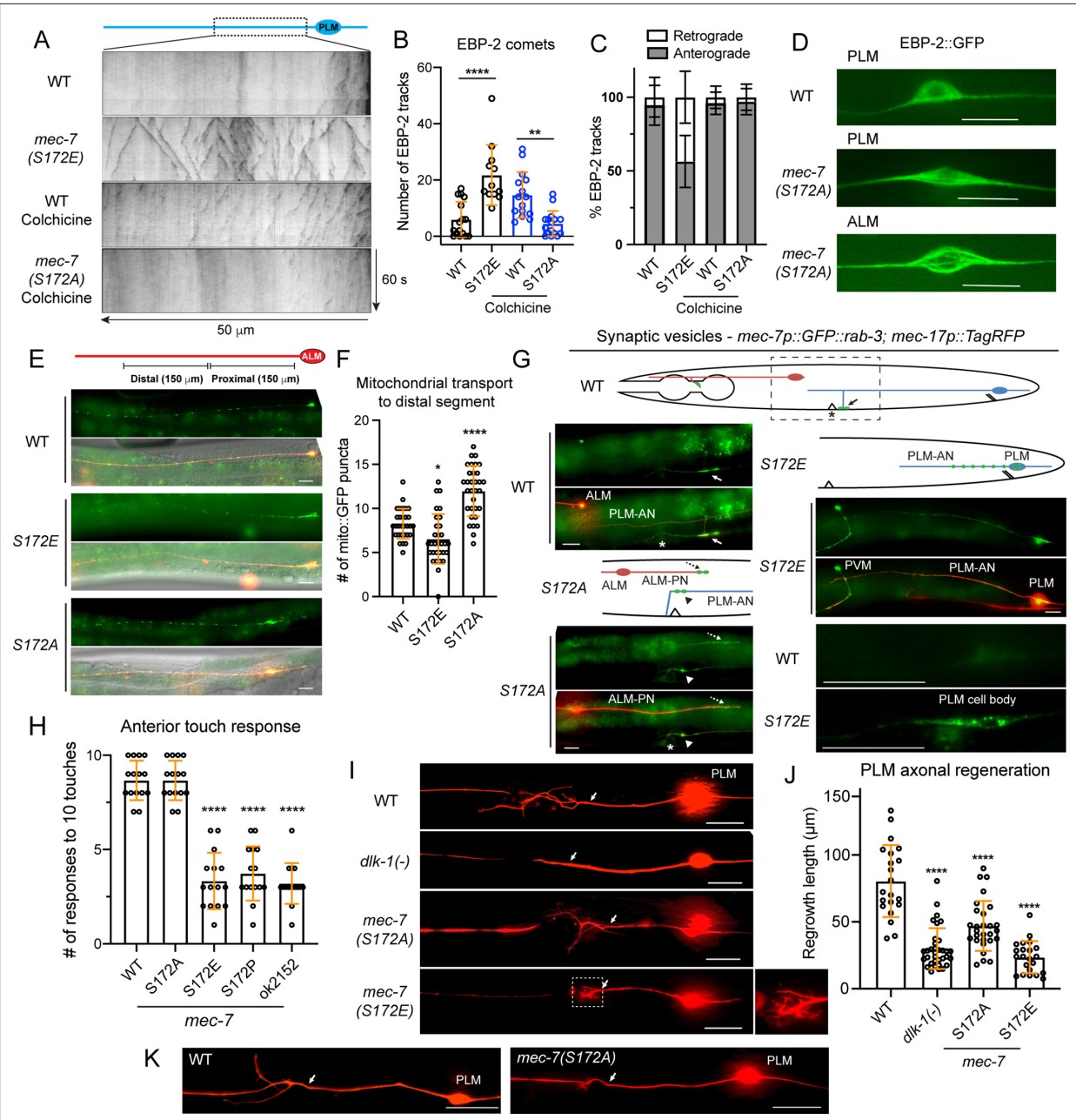

**Figure 2.** MEC-7/β-tubulin S172 phosphorylation regulates MT dynamics, cargo transport, mechanosensory function, and axonal regeneration.
(**A**) Representative kymographs of EBP-2::GFP dynamics in the PLM-AN of various strains. The wild-type animals and *S172A* mutants were subjected to a mild colchicine treatment to increase MT dynamics and imaged after a one-hour recovery. (**B**) Quantification of the number of EBP-2 tracks. One, two, and four asterisks indicate p<0.05, 0.01, and 0.0001, respectively, in statistical significance when compared with the wild type. (**C**) Percentages of retrograde and anterograde movement for the EBP-2 comets. (**D**) Comparison of the EBP-2::GFP signal in the cell body of wild-type and *mec-7(S172A)* animals. (**E**) Distribution of mitochondria in the ALM-AN indicated by the *jsIs609 [mec-7p::mitoGFP]* signal. (**F**) The quantification of the number of mitoGFP puncta in the distal segment of ALM-AN (150–300 μm away from the cell body). (**G**) Localization of the synaptic vesicles (GFP::RAB-3), which are indicated by green dots in the cartoon of S172 mutants. The fluorescent image for the wild type is a representative image of the region in the dashed box of the cartoon; arrows indicate the normal localization of RAB-3 signal to the synapses made by synaptic branch of PLM-AN in the ventral nerve cord posterior to the vulval position (indicated by the asterisk). In S172A mutants, RAB-3 is mistargeted to the ALM-PN (dashed arrow) and accumulates at the PLM-AN (arrowhead) when the synaptic branch fails to extend, and the axon hooks ventrally. In S172E mutants, RAB-3 signal was trapped in the cell body or the proximal segment of the neurite as PLM-AN is severely shortened. (**H**) Anterior touch responses of various *mec-7* mutants; *mec-7(ok2151)* is a deletion allele. (**I**) Representative axonal regrowth of PLM-AN following laser axotomy in various strains; arrows indicate the cut site. (**J**) Quantification of PLM regrowth length for the regrowth cases. (**K**) Examples of reconnections after laser axotomy; arrows indicate the cut site. Scale bars = 20 μm for all panels.

*Figure 2 continued on next page*

*Figure 2 continued*

The online version of this article includes the following source data for figure 2:

**Source data 1.** Numeric data for *Figure 2B–C, F, H and J*.

in *mec-7(S172A)* mutants (*Figure 2C*). These results supported that β-tubulin S172 phosphorylation promotes dynamic MTs, which negatively regulates neurite growth.

In addition, the EBP-2::GFP signal in the cell body showed an unusual fiber-like shape in the *mec-7(S172A)* mutants instead of the diffusive pattern in the wild-type (*Figure 2D*), indicating the possible existence of MT bundles in the *S172A* cell body. In the wild-type TRNs, the MT bundles were only found in the axons but not the cell body according to previous electron microscopy studies (*Chalfie and Thomson, 1979*). Another hypothesis is that the incorporation of nonphosphorylatable MEC-7(S172A) into MTs might affect GTP hydrolysis and thus expand the EBP-2 binding region beyond the MT tips, since EB2 has a specific nucleotide-dependent binding property (*Roth et al., 2018*).

S172 phosphorylation also affected cargo transport in TRNs. Mitochondrial transport was impaired in *mec-7(S172E)* mutants, resulting in lower density of mitochondria along the axons (*Figure 2E–F*). In contrast, *mec-7(S172A)* mutants showed higher accumulation of mitochondria in the distal segment of the ALM-AN than the wild type. Moreover, the transport of synaptic vesicles was also disrupted in *mec-7(S172E)* mutants. For example, wild-type PLM-AN branches at a position close to the vulva; the branch forms chemical synapses with the neurites in the ventral nerve cord, where the synaptic vesicles are transported to (*Figure 2G*). In *mec-7(S172E)* mutants, the vesicles were mostly trapped in the cell bodies or along the proximal segment of the neurite, probably due to highly unstable MTs (*Figure 2G*). About 40% of PLM-AN in *mec-7(S172A)* mutants grew no branches or had a 'hook' defect, and in these cases synaptic vesicles were trapped at the distal end of the neurite (*Figure 2G*). Interestingly, synaptic vesicles were also transported to the ectopic ALM-PN in *mec-7(S172A)* mutants (*Figure 2G*), as previously observed in other *mec-7(neo)* mutants (*Zheng et al., 2017*).

At the behavioral level, both S172E and S172P mutants showed strong defects in touch sensitivity, similar to the *mec-7(-)* deletion allele. The S172A mutants, however, did not reduce touch sensitivity (*Figure 2H*). Since stable MTs are essential for the mechanosensory function of the TRNs (*Savage et al., 1989*), reducing MT stability through S172 hyperphosphorylation was expected to affect TRN functions.

## Optimal level of MEC-7/β-tubulin S172 phosphorylation is required for axonal regeneration

One of the important functions of MTs is to support neuronal regeneration, and previous work have established the TRNs as an important model for axonal regeneration (*Wu et al., 2007*). In our hands, following laser axotomy, ~72% wild-type PLM neurons were able to regrow the proximal axon for a short length to connect with the distal fragment (defined as reconnection), whereas the other ~28% PLM did not reconnect the severed axons and instead grew the proximal axon for a substantial length (defined as regrowth). We found that the *mec-7(S172E)* mutants were not able to reconnect the severed axons (only 4% reconnection) because the regrowth length was very short (*Figure 2I–K*). A mesh-like structure with short sprouts were observed at the regrowth site. We reasoned that the dynamic MTs may be too unstable to support any substantial regrowth in the S172E mutants. Notably, these regeneration defects were different from that seen in the *dlk-1(-)* mutants, which showed significantly reduced regrowth but had no extensive sprouting (*Figure 2I*) probably due to the lack of MT polymerization at the injury site (*Ghosh-Roy et al., 2012*). DLK-1 is a dual leucine zipper MAPKKK essential for MT growth in regeneration. The phenotypic differences between *dlk-1(-)* and *mec-7(S172E)* mutants indicated different underlying mechanisms.

The axonal regeneration in *mec-7(S172A)* mutants was also different from the wild-type. Although 40% of the animals were able to reconnect the proximal axons with the distal segment, the regrowth length was shorter than the wild-type animals (*Figure 2I–J*). Moreover, a highly branched tree-like structure was seen at the regrowth site. It appears that the regrowth had no clear direction. The regrowing axon extended multiple short processes towards different directions, but none of these processes developed into a prominent axon. One possible explanation is that the hyperstable MTs led to excessive neurite outgrowth and failure to properly respond to guidance cues during regeneration.

Interestingly, another *mec-7* neomorphic allele that caused a P220S substitution also showed defects in axonal regeneration due to hyperstable MTs (*Kirszenblat et al., 2013*). Thus, the above results suggested that an optimal level of β-tubulin S172 phosphorylation is required for effective axonal regeneration.

## Enzymes mediating S172 phosphorylation in *C. elegans*

We next searched for the enzyme that mediates MEC-7 S172 phosphorylation by testing the homologs of known kinases involved in phosphorylating β-tubulin. We first tested the two close homologs of Minibrain kinase in *C. elegans*, *mbk-1* and *mbk-2*, both of which were expressed in the TRNs based on fluorescent reporters (*Figure 3—figure supplement 1A–B*). Mutations in *mbk-2* led to embryonic lethality, and only a few animals could hatch then arrest at early larval stages (*Quintin et al., 2003*). We examined these arrested larvae did not observe excessive neurite growth as in S172A mutants. *mbk-1(-)* mutants were viable but did not show any TRN developmental defects. We also examined the arrested larvae in *mbk-1(-) mbk-2(-)* double mutants and could not observe the ectopic ALM-PN either (*Figure 3—figure supplement 1D*). To knock down *mbk-2* expression in the TRNs specifically, we expressed dsRNA against *mbk-2* from a TRN-specific *mec-17* promoter in both wild-type and *mbk-1(-)* background and again failed to detect any changes in TRN morphology. We then attempted to degrade the MBK-2 proteins specifically in the TRNs using a ZF1/ZIF-1 system (*Wang et al., 2017*). We first edited the endogenous *mbk-2* locus to fuse GFP to an exon shared by all *mbk-2* isoforms; to our surprise, we could not detect MBK-2::GFP signal in the TRNs although we could observe the expected embryonic expression (*Figure 3—figure supplement 2A–B*). We reasoned that the endogenous MBK-2 level might be very low in the TRNs. We then crossed *mbk-2::gfp* into a strain expressing the GFP nanobody::ZIF-1 fusion specifically in the TRNs; the resulted animals did not show long ALM-PN, suggesting that MBK-2, even if expressed at low level, was not likely to regulate TRN morphogenesis.

Minibrain kinase has a third, distant homolog in *C. elegans*, *hpk-1*, which is also the homolog of human homeodomain-interacting protein kinase 1 (HIPK1). *hpk-1* was expressed in the TRNs, but *hpk-1(-)* mutants did not show defects in TRN development (*Figure 3—figure supplement 1C–D*). Based on the above results, we concluded that either high level of redundancy existed among the Minibrain homologs or the Minibrain kinase was not the enzyme catalyzing β-tubulin S172 phosphorylation in *C. elegans*.

Interestingly, the loss of the cyclin-dependent kinase Cdk1 led to the growth of ectopic ALM-PN, similar to *mec-7(S172A)* mutants. We analyzed two deletion alleles (*he5* and *ok1882*) of *cdk-1* (*C. elegans* Cdk1), both of which caused lethality. Maternally rescued homozygous *cdk-1(-)* animals were arrested in early larval stages, but some escapers could reach late larval stages. We found that the arrested *cdk-1(he5)* and *cdk-1(ok1882)* larvae grew an ectopic ALM-PN (*Figure 3A–B*). Because these animals could not grow to the adult size, we measured the ALM-PN/ALM-AN ratio and found that *cdk-1(-)* mutants had a similar ratio as the *mec-7(S172A)* animals (*Figure 3C*). Moreover, the *cdk-1(-); mec-7(S172A)* double mutants showed the ALM-PN length comparable to the two single mutants (*Figure 3D*), suggesting that they act in the same pathway.

Finally, we tested the enzymatic activity of CDK1 towards the recombinant MEC-12/MEC-7 α/β-tubulin heterodimer and found that CDK1 could indeed phosphorylate the S172 of MEC-7/β-tubulin (*Figure 3E–F*). In contrast, recombinant MBK-2 proteins did not show kinase activities towards S172 of MEC-7 either in the same MEM reaction buffer (*Figure 3E*) or in the BRB80 buffer (*Figure 3—figure supplement 3*) used by the previous studies of *Drosophila* Minibrain (*Ori-McKenney et al., 2016*), which was consistent with the lack of ALM-PN phenotype in *mbk-2(-)* mutants. Although *cdk-1/Cdk1* was known to be expressed mostly in mitotic cells, some studies suggested a role for Cdk1 in postmitotic neurons (*Kim and Bonni, 2008*; *Yuan et al., 2008*), and cell cycle regulators (including cyclins) showed expression in neurons (*Akagawa et al., 2021*). In fact, single-cell transcriptomic studies found weak expression of cyclin A (*cya-1*) and cyclin B (*cyb-1*) in ALM neurons (*Taylor et al., 2021*). Thus, Cdk1 may regulate neuronal differentiation by catalyzing β-tubulin S172 phosphorylation. Nevertheless, we could not rule out the possibility that other kinases may also regulate S172 phosphorylation.

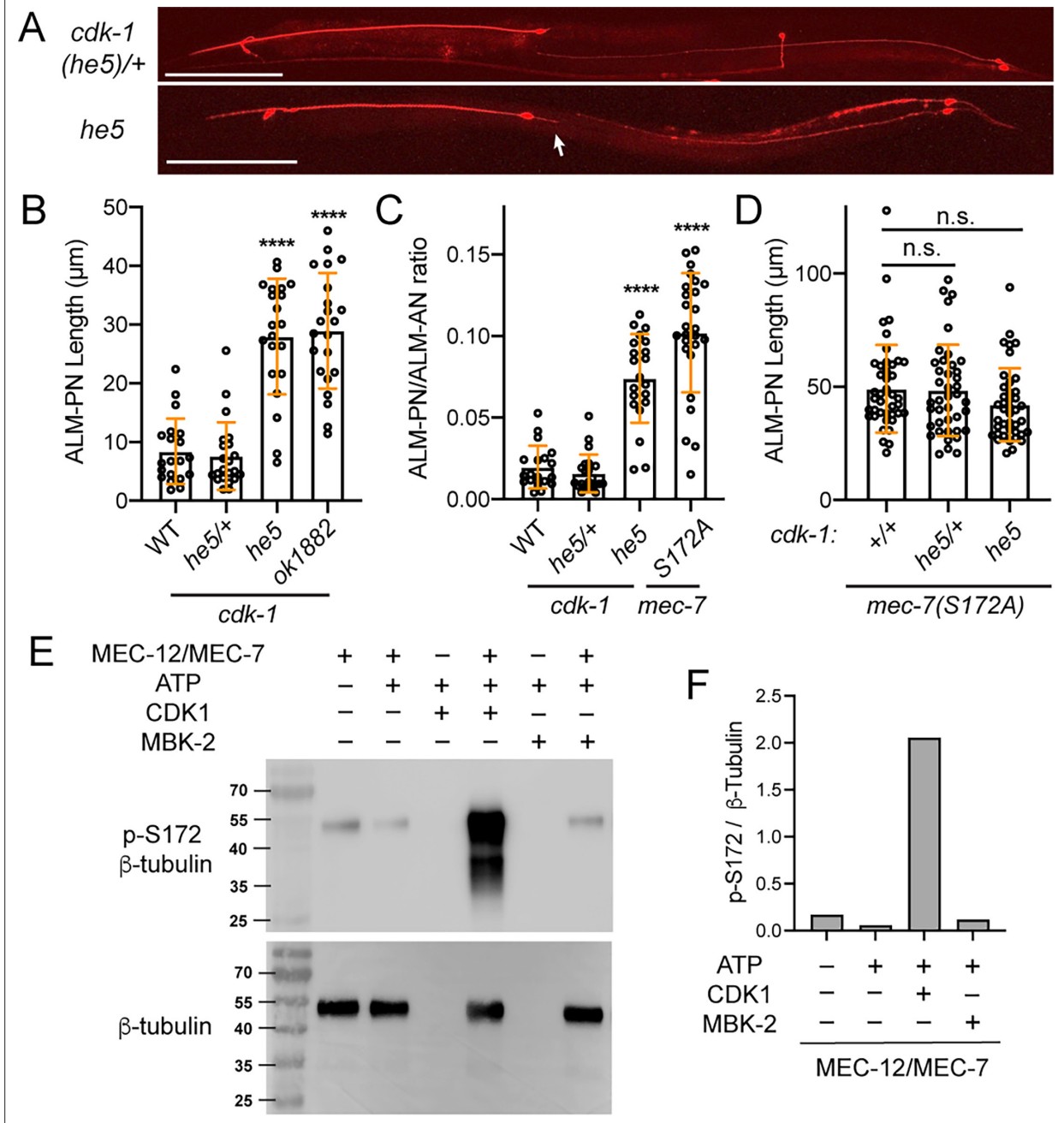

**Figure 3.** CDK-1 mediates MEC-7/β-tubulin S172 phosphorylation and regulates neurite growth. (**A**) Images of TRN morphologies in *cdk-1(he5)/+* heterozygotes and *cdk-1(he5)* homozygotes (escapers). Arrow points to the ectopic ALM-PN. Scale bar = 100 μm. (**B**) The length of ALM-PN in wild-type and *cdk-1* mutants. ALM-PN length in the escapers of *he-5* and *ok1882* homozygous mutants were measured. (**C**) The ratio of ALM-PN length to ALM-AN length in various strains. (**D**) The ALM-PN length in animals carrying both *mec-7(S172A)* and *cdk-1* mutations. (**E**) Western blot of the kinase reaction samples using anti-phospho-S172 (p–S172) antibodies. The kinase reaction was conducted in the MEM buffer with 475 nM CDK1/Cyclin B or 500 nM MBK-2m. The weak p-S172 signal in the first lane likely resulted from the nonspecific interaction of the anti-p-S172 antibodies with the unmodified MEC-12/MEC-7 heterodimer or the background phosphorylation level of the recombinant dimer. Anti-β-tubulin antibodies were used to ensure the equal loading of the tubulin dimers. (**F**) Quantification of the grey intensity ratio of p-S172 to β-tubulin signals in the western blot.

The online version of this article includes the following source data and figure supplement(s) for figure 3:

Source data 1. Numeric data for *Figure 3B–D , and F*.

Source data 2. Labeled uncropped western blot image for the blots shown in *Figure 3E* and *Figure 3—figure supplement 3*.

Source data 3. Unlabeled uncropped western blot image for the blots shown in *Figure 3E* and *Figure 3—figure supplement 3*.

*Figure 3 continued on next page*

*Figure 3 continued*

**Figure supplement 1.** Expression and phenotypes of the minibrain homologs, *mbk-1*, *mbk-2*, and *hpk-1* in the TRNs.

**Figure supplement 1—source data 1.** Numeric data for *Figure 3—figure supplement 1D*.

**Figure supplement 2.** Potential TRN-specific degradation of MBK-2 does not result in the growth of long ALM-PN.

**Figure supplement 2—source data 1.** Numeric data for *Figure 3—figure supplement 2C*.

**Figure supplement 3.** Kinase assay for MBK-2 in BRB80 buffer.

## MEC-12/α-tubulin K40 acetylation is not essential for TRN morphology and function

α-tubulin acetylation, which occurred at the lysine 40 (K40) in the MT lumen, has been a marker for stable MTs, particularly in neurons. Early in vitro studies found that tubulin acetylation did not affect the kinetics of MT polymerization (*Maruta et al., 1986*), while recent work showed that K40 acetylation may reduce lateral contact between protofilament, which increased the flexibility of MTs and provided resistance to mechanical stress (*Portran et al., 2017*). In neurons, tubulin acetylation promotes the transport of vesicles along the axons (*Even et al., 2019*). Altered tubulin acetylation levels were often correlated with defects in neuronal morphogenesis (*Chang et al., 2009*), but whether K40 acetylation has a direct impact on axonal growth is less clear.

Although K40 is quite conserved in α-tubulins across species, among the nine α-tubulin isotypes in *C. elegans*, only MEC-12 has a K40 residue (*Figure 4—figure supplement 1A*). We created acetyl-mimicking K40Q and unmodifiable K40R mutants through gene editing and confirmed the change of tubulin acetylation status by observing the loss of anti-acetyl-α-tubulin (K40) antibody staining signal in both K40R and K40Q mutant animals (*Figure 4—figure supplement 2A*), which is consistent with previous observations in *Drosophila* (*Mao et al., 2017*). Neither *mec-12(K40R)* nor *mec-12(K40Q)* mutations generated significant defects in TRN neurite morphogenesis (*Figure 4A–B*). The only notable defect was that ~20% of PLM-AN failed to produce the synaptic branch in *mec-12(K40Q)* mutants (*Figure 4C*), suggesting that K40 hyperacetylation might affect MT dynamics during neurite branching. Moreover, *mec-12(K40Q)* mutants showed moderately reduced regrowth of PLM-AN after laser axotomy (*Figure 4D*), suggesting that K40 hyperacetylation might also affect MT functions during axonal regeneration.

Previous work found that the α-tubulin acetyltransferase MEC-17 plays an important role in controlling MT number and organization and neurite growth (*Cueva et al., 2012*; *Topalidou et al., 2012*). However, no significant downregulation of MEC-12 K40 acetylation was observed in *mec-17(-)* mutants due to the redundancy with another acetyltransferase ATAT-2 (*Akella et al., 2010*; *Topalidou et al., 2012*), suggesting that the phenotypes of *mec-17(-)* mutants was not due to the loss of K40 acetylation. Moreover, *Topalidou et al., 2012* found that expression of the enzymatically inactive MEC-17(G121W and G123W; termed as dW) mutants rescued some TRN defects of *mec-17(-)* mutants, including the touch sensitivity and the growth of ectopic ALM-PN, whereas *Shida et al., 2010* found that the enzymatically dead MEC-17 could not rescue the touch defects. Both studies used exogenous transgenic expression of MEC-17 mutants, whose expression levels were uncertain.

We created the *mec-17(dW)* allele by editing the endogenous *mec-17* locus and confirmed that MEC-12 K40 acetylation was not significantly reduced upon the loss of enzymatic activity of MEC-17 in *mec-17(dW)* mutants but was completely lost in *mec-17(dW); atat-2(-)* double mutants (*Figure 4—figure supplement 2B–C*). Importantly, the defects in touch sensitivity found in *mec-17(-)* mutants was not observed in *mec-17(dW)* mutants (*Figure 4E*), supporting that MEC-17 has separate functions independent of its acetyltransferase activity. This result is consistent with the normal touch sensitivity in *mec-12(K40R)* mutants (*Figure 4E*) and indicates that K40 acetylation is not required for TRN mechanosensation.

Furthermore, compared to *mec-17(-)* mutants, the ectopic growth of ALM-PN and the neurite swelling and looping phenotypes, which were caused by MT bending in the absence of MEC-17 (*Topalidou et al., 2012*), were partially rescued in *mec-17(dW)* mutants (*Figure 4F–H*). These results supported a non-enzymatic function of MEC-17 in regulating MT organization and neurite outgrowth. Importantly, none of the remaining *mec-17(dW)* defects were found in *mec-12(K40R)* mutants, suggesting that even the enzymatic activity of MEC-17 may regulate neurite morphology through an α-tubulin K40-independent mechanism, which may involve the acetylation of other residues of

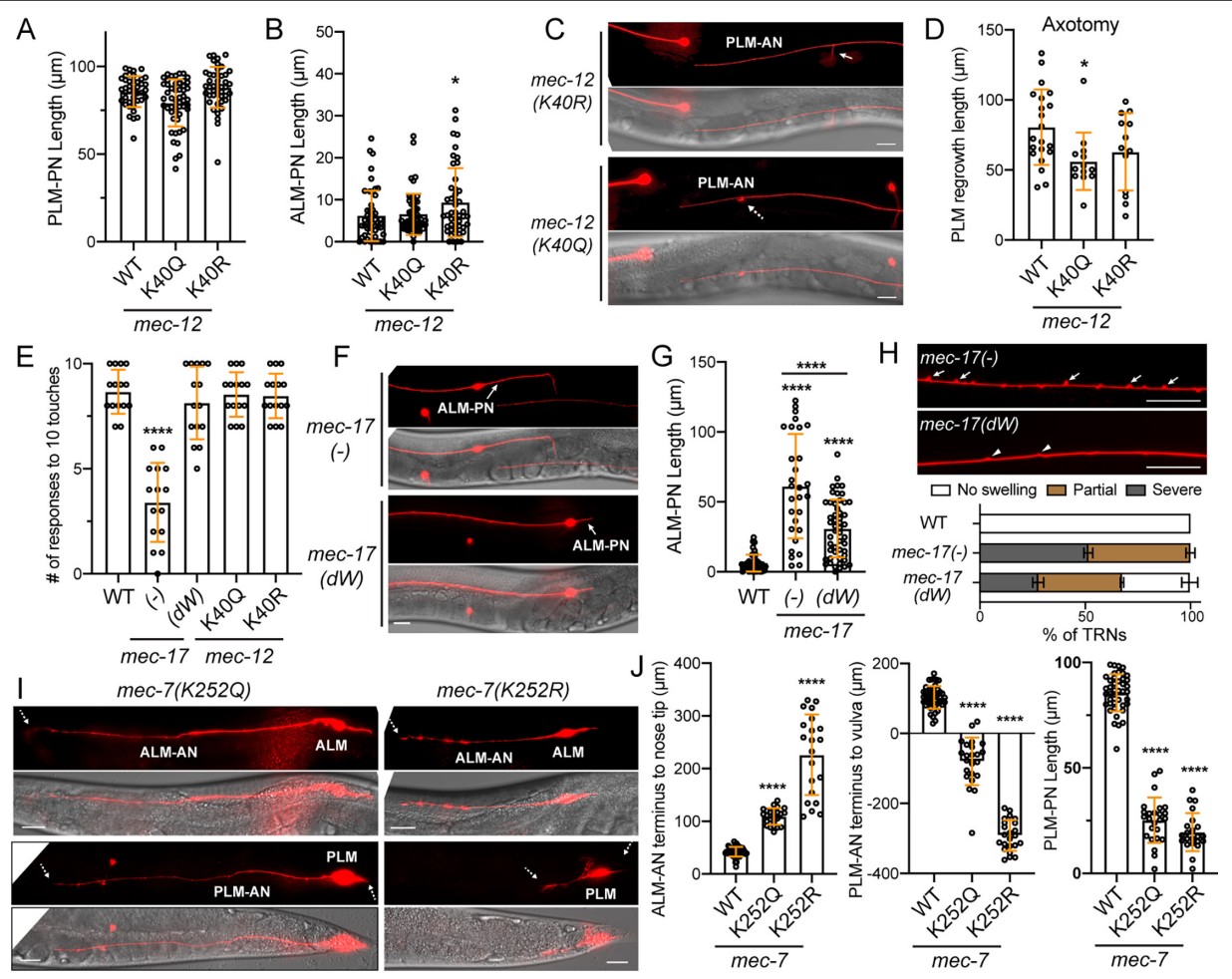

**Figure 4.** The effects of tubulin acetylation on neurite development. (**A**) The PLM-PN length in *mec-12* K40 mutants. (**B**) The length of ALM-PN in *mec-12* K40 mutants. (**C**) 20% of PLM-AN failed to extend the synaptic branch in *mec-12(K40Q)* mutants. Arrow points to the normal synaptic branch in *mec-12(K40R)* mutants and the dashed arrow indicates the branching defects. See *Figure 1H* for the wild-type control. (**D**) Quantification of the PLM regrowth length after laser axotomy in *mec-12* K40 mutants. (**E**) Anterior touch responses of *mec-17* and *mec-12* mutants. The deletion allele *ok2109* was used for *mec-17(-)* and the *unk126* (G121W & G123W) allele was used for *mec-17(dW)*. (**F**) Comparison of ALM-PN in *mec-17(-)* and *mec-17(dW)* mutants. (**G**) Quantification of ALM-PN lengths in *mec-17* mutants. Four asterisks indicate p<0.0001 in a post-ANOVA Tukey's honestly significant difference (HSD) test. (**H**) Comparison and quantification of the neurite swelling and looping phenotypes in *mec-17(-)* and *mec-17(dW)* mutants (ALM-AN is shown). For the quantification, severe swelling (arrows) is defined by having two or more bumps taller than 1 μm within 100 μm length of axon; partial swelling (arrowheads) is defined by having smaller or fewer bumps than the severe ones; no swelling is defined by having no >0.5 μm bumps in the entire axon. (**I**) The ALM and PLM morphologies in *mec-7* K252 mutants. Dashed arrows point to the termini of the shortened ALM-AN or PLM-AN or PLM-PN. See *Figure 1B* for the wild-type control. (**J**) Quantification of the shortening of the three neurites. For ALM-AN, the distance from the neurite ending to the nose tip was measured; the bigger the gap, the shorter the ALM-AN. For PLM-AN, the distance from the neurite terminus to the vulva was measured. Positive value means the neurite grew past the vulva and negative value means the neurite did not reach the vulva. Scale bars = 20 μm for all panels.

The online version of this article includes the following source data and figure supplement(s) for figure 4:

**Source data 1.** Numeric data for *Figure 4A–B, D–E, G–H , and J*.

**Figure supplement 1.** Evolutionary conservation of α-tubulin K40 and β-tubulin K252 residues.

**Figure supplement 2.** *mec-12* K40R and K40Q mutations and the combined loss of *mec-17* and *atat-2* affect α-tubulin K40 acetylation.

α-tubulin or other proteins. Removing *atat-2* in the *mec-17(-)* and *mec-17(dW)* mutants did not make the neurite morphological defects worse despite eliminating K40 acetylation.

## Mutation of MEC-7/β-tubulin K252 strongly impairs neurite development

The lack of obvious phenotypes in K40 mutants prompted us to look for other potential tubulin acetylation sites, such as the β-tubulin K252, which was reported to be acetylated by San/Nat5 in human cells (*Chu et al., 2011*). K252 and flanking residues are highly conserved among β-tubulin isotypes and across species (*Figure 4—figure supplement 1B*). We created acetyl-mimicking K252Q and unmodifiable K252R mutants by editing the endogenous *mec-7* locus and found that both mutations caused strong neurite growth defects, similar to previously categorized *mec-7(anti)* mutants (*Figure 4I–J*). All TRN neurites were severely shortened in these mutants with K252R showing a stronger phenotype than K252Q mutants. Previous work suggested that acetylation of K252 reduced the incorporation of tubulin into MTs by neutralizing the positive charge of K252, which is located at the intradimer interface of α/β-tubulins (*Chu et al., 2011*). Our findings with the acetyl-mimicking K252Q mutants supported this hypothesis, but the K252R results are rather unexpected, as this mutation in theory would not affect the amino acid charge at that position. However, we could not rule out the possibility that substituting lysine with arginine caused conformational change of the tubulin heterodimer, creating dominant-negative effects. Deletion of the *C. elegans* homolog of San/Nat5, *F40F4.7(tm2414)*, did not cause any TRN defects. Thus, it remains unclear what enzyme mediates the K252 acetylation of MEC-7 in *C. elegans*.

## Polyglutamylation on the C-terminal tail of tubulins reduces MT stability and prevents excessive neurite growth in TRNs

The C-terminal tail of the tubulins harbors the sites for polymodifications, such as polyglutamylation and polyglycylation, which occur on the glutamic acid residues and regulate the interaction between MTs and various motor proteins and MAPs. Polyglutamylation is associated with stable MTs in cilia, centrioles, mitotic spindles, and axons (*Kubo et al., 2010*; *Lacroix et al., 2010*; *Suryavanshi et al., 2010*), while polyglycylation marks the stable axonemal MTs in flagellated and ciliated cells (*Xia et al., 2000*). In *C. elegans*, polyglutamylation regulates ciliary microtubule organization in ciliated sensory neurons, but no polyglycylation signal was observed due to the lack of polyglycylase homologs (*Kimura et al., 2010*).

The MEC-12 α-tubulin C-terminal tail consists of 15 amino acids (436-450aa; GVDSMEDNGEEGDEY), and the last 12 (439-450aa) showed significant sequence divergence among the isotypes (*Figure 5—figure supplement 1*). To understand the function of the C-terminal tail, we first deleted the DNA encoding the last seven amino acids (ΔGEEGDEY) in the endogenous *mec-12* locus and found that the mutants grew an ectopic ALM-PN (*Figure 5A*), suggesting that the overall effects of this fragment may be MT-destabilizing and thus their removal induced excessive neurite growth. Changing all seven amino acids to alanine resulted in similar phenotypes. There are four glutamic acid residues in MEC-12 C-terminal tail and three of them are in the last seven amino acids; these residues are potential sites for polyglutamylation. Mutating any of them to alanine could induce the growth of ALM-PN and converting all to alanine produced the strongest phenotype, suggesting that their functions are additive (*Figure 5A–B*). Deleting the glutamic acid residues produced similar effects as changing them to alanine. Therefore, we hypothesized that disabling tubulin polyglutamylation increased MT stability, resulting in the growth of an extra neurite. Indeed, the *mec-12(4Es-A)* mutants, in which all four glutamic acids were mutated to alanine, showed significantly reduced number of EBP-2 tracks in sensitized conditions (*Figure 5C–D*), indicating reduced MT dynamics upon the loss of polyglutamylation. Nevertheless, the MT polarity was preserved in the *mec-12(4Es-A)* mutants (*Figure 5E*).

Previous studies found that tubulin polyglutamylation may inhibit axonal regeneration by affecting MT dynamics and the removal of the *ttll-5* gene that codes for a tubulin tyrosine ligase-like protein responsible for polyglutamylation enhanced regeneration (*Ghosh-Roy et al., 2012*). We found that the *mec-12(4Es-A)* mutants also had enhanced axonal regeneration, which is similar to the *ttll-5(-)* mutants (*Figure 5F*). Importantly, the *mec-12(4Es-A); ttll-5(-)* double mutants did not show a phenotype

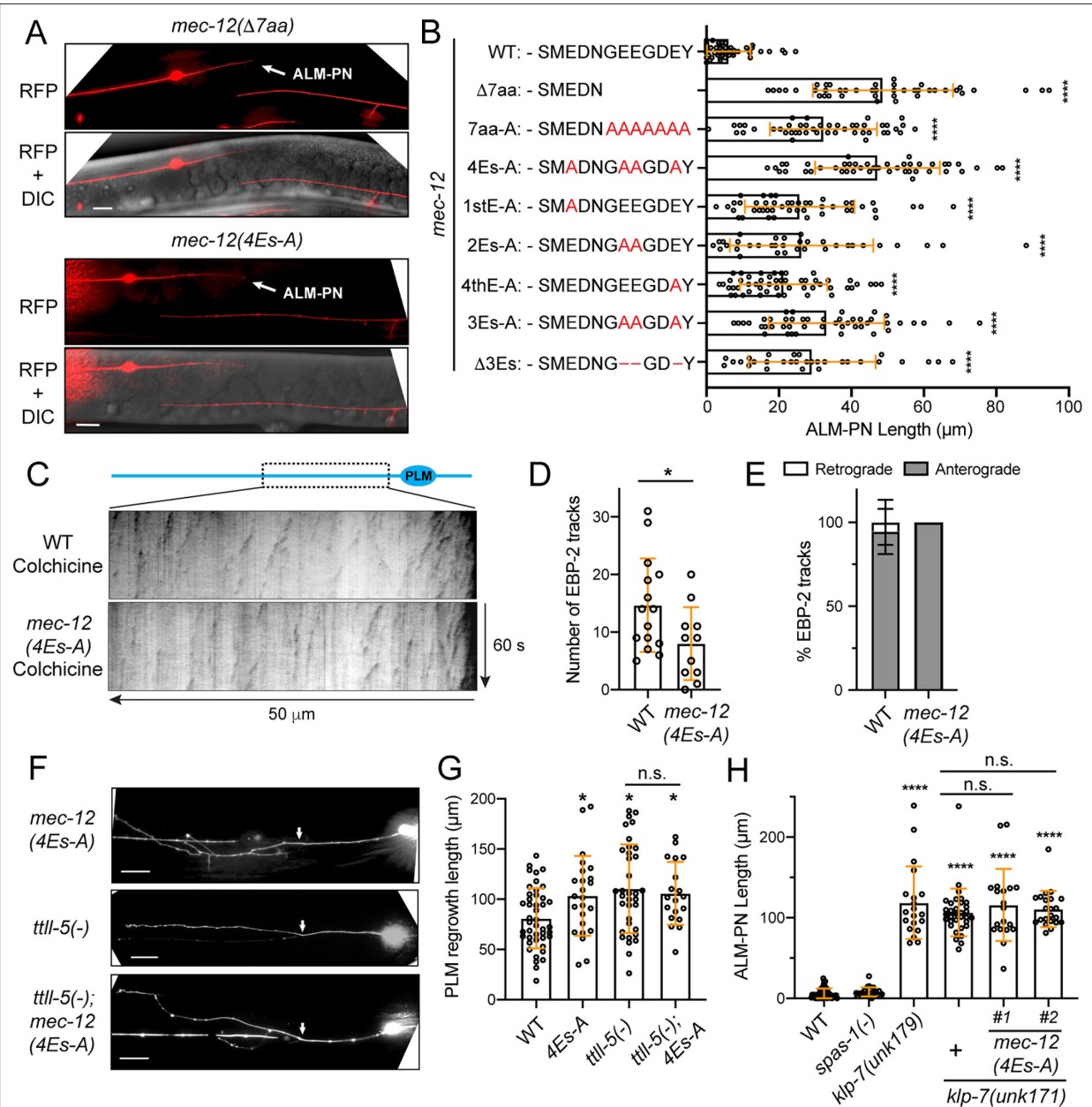

**Figure 5.** Tubulin polyglutamylation regulates neurite growth and regeneration. (**A**) The growth of ectopic ALM-PN (arrow) in *mec-12* mutants lacking the polyglutamylation sites. (**B**) The quantification of ALM-PN length in various *mec-12* mutant strains. The changes made to the last twelve amino acids of MEC-12 in the various alleles are shown. (**C**) Kymographs showing EBP-2::GFP dynamics in wild-type and *mec-12(4Es-A)* animals after a mild colchicine treatment and a one-hour recovery. (**D**) Quantification of the number of EBP-2 comets in *mec-12(4Es-A)* mutants. (**E**) The percentage of retrograde and anterograde EBP-2 movements. (**F**) PLM regrowth following laser axotomy in *mec-12(4Es-A)* and *ttll-5(tm3360)* mutants and the double mutants; arrows indicate the cut site. (**G**) The quantification of PLM regrowth length in various strains. (**H**) ALM-PN lengths in *spas-1(tm683)*, *klp-7(unk179; Δ11bp)*, *klp-7(unk171; Δ14bp)*, and *klp-7; mec-12(4Es-A)* double mutants. *unk171* was generated by gene editing in both the wild-type and the *mec-12(4Es-A)* background, and #1 and #2 were two independent lines. See *Figure 5—figure supplement 2* for the details of the *klp-7* alleles. Scale bars = 20 μm for all panels; n.s. indicates no statistical significance in a post-ANOVA Tukey's HSD test.

The online version of this article includes the following source data and figure supplement(s) for figure 5:

**Source data 1.** Numeric data for *Figure 5B, D–E , and G–H*.

**Figure supplement 1.** Sequence divergence of α-tubulin C-terminal tails in *C. elegans*.

**Figure supplement 2.** Mutations in *klp-7* result in the growth of an ectopic ALM-PN.

**Figure supplement 3.** The lack of cargo transport and mechanosensory defects in *mec-12(4Es-A)* mutants.

**Figure supplement 3—source data 1.** Numeric data for *Figure 5—figure supplement 3C–D*.

stronger than either of the two single mutants (*Figure 5G*), suggesting that the major function of the four glutamic acid residues is mediated by polyglutamylation.

Tubulin polyglutamylation was found to stimulate spastin-mediated MT severing in human cells (*Lacroix et al., 2010*). We found, however, that the loss of the *C. elegans* homolog spastin, *spas-1*, did not produce the ALM-PN phenotype (*Figure 5H*), suggesting that spastin may not mediate the effects of polyglutamylation in the TRNs. On the other hand, the MT-depolymerizing kinesin-13 homolog *klp-7* was found to mediate the effects of polyglutamylation in regeneration (*Ghosh-Roy et al., 2012*). In fact, we and others have previously found that the loss of *klp-7* led to the growth of an ectopic ALM-PN similar to the *mec-12(4Es-A)* mutants (*Puri et al., 2021*; *Zheng et al., 2017*). In this study, we created *klp-7(-); mec-12(4Es-A)* double mutants by inactivating *klp-7* in *mec-12(4Es-A)* animals through Cas9-mediated gene editing since the two genes are on the same chromosome (*Figure 5—figure supplement 2*). We found that the *klp-7(-); mec-12(4Es-A)* double mutants did not enhance the ALM-PN phenotype of the *klp-7(-)* single mutants (*Figure 5H*), suggesting that MEC-12 polyglutamylation likely acted through KLP-7. Thus, we suspect that tubulin polyglutamylation of the α-tubulin C-terminal tail might enhance the binding of KLP-7 to MTs to increase MT dynamics.

Despite the above defects, tubulin polyglutamylation did not affect the transport of cargos, such as the synaptic vesicles and mitochondria, along the axons (*Figure 5—figure supplement 3A–C*). Unlike the *mec-7(S172A)* mutants, which also grew an ectopic ALM-PN, *mec-12(4Es-A)* animals did not show

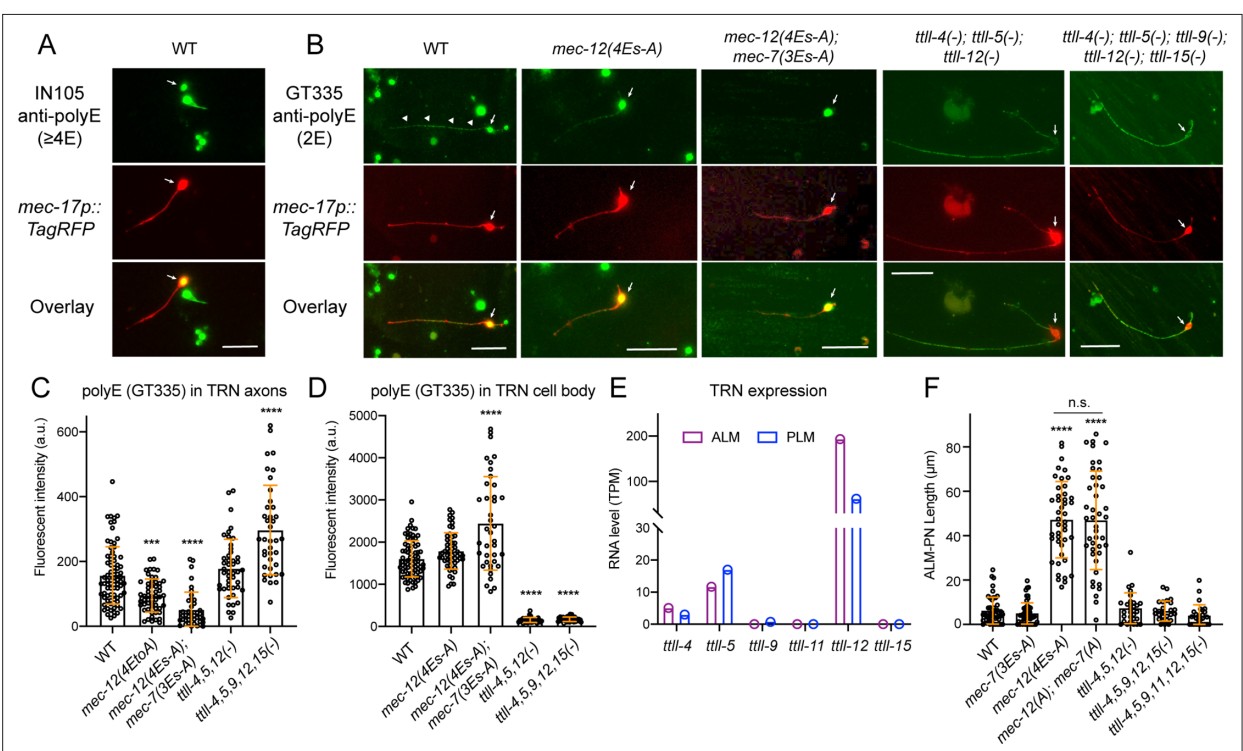

**Figure 6.** Substitution of glutamates in MEC-12 and MEC-7 C-terminal tails eliminate tubulin polyglutamylation in the axons. (**A**) Antibody staining of in vitro cultured TRNs (labeled by RFP) with IN105 anti-polyglutamylation antibodies that recognize a chain of four or more glutamates. (**B**) Antibody staining of in vitro cultured TRNs from various strains with GT335 anti-polyglutamylation antibodies that recognize the branchpoint and a side chain of two glutamyl units. CGZ1554 *ttll-4(tm3310); ttll-5(tm3360); ttll-12(unk185); uIs115* and CGZ1475 *ttll-4(tm3310); ttll-5(tm3360) ttll-9(tm3889) ttll-15 (tm3871); ttll-12(unk185); uIs115* were used in the experiment. Arrows point to the TRN cell bodies and arrowheads point to the staining signal in the axon. (**C–D**) Quantification of the fluorescent intensity (in arbitrary units or a.u.) for the polyE staining of TRN axons and cell bodies using GT335. (**E**) RNA level of the *ttll* genes in ALM and PLM neurons according to the L4 stage single-cell transcriptomic data obtained from https://wormbase.org/. (**F**) ALM-PN length in *mec-7*, *mec-12*, and *ttll* mutant strains. CGZ1554, CGZ1475, and CGZ1474 *ttll-4(tm3310); ttll-11(tm4059); ttll-5(tm3360) ttll-9(tm3889) ttll-15(tm3871); ttll-12(unk185); zdIs5* were used. Scale bars = 20 μm for all panels. Three and four asterisks indicate p<0.001 and 0.0001, respectively, in a post-ANOVA Dunnett's test comparing all mutants with the wild-type animals.

The online version of this article includes the following source data and figure supplement(s) for figure 6:

**Source data 1.** Numeric data for *Figure 6C–F*.

**Figure supplement 1.** Sequence divergence of the C-terminal tail among β-tubulin isotypes and the molecular details of *ttll-12* CRISPR allele.

the transport of cargos into the extra ALM-PN (compare *Figure 5—figure supplement 3A–C* with *Figure 2E and G*). These results suggest that the MT defects caused by the lack of polyglutamylation may be more subtle than the general alteration of MT dynamics in S172 mutants. Moreover, the touch sensitivity of the *mec-12(4Es-A)* mutants also appeared to be normal, suggesting that polyglutamylation may not be essential for the mechanosensory function of TRNs (*Figure 5—figure supplement 3D*).

## Polyglutamylation in the TRN neurites are abolished by mutations in *mec-12* and *mec-7*

To confirm that the above mutations affected polyglutamylation, we stained the in vitro cultured TRNs with two anti-polyglutamylation antibodies, GT335 and IN105. The former recognized the octapeptide EGEGE(*EE)G containing a branchpoint and a side chain of two glutamyl units (*Wolff et al., 1992*), and the latter recognized a linear chain of four or more glutamates (*Rogowski et al., 2010*). Interestingly, IN105 only stained the cell body of TRNs, while GT335 stained both the cell body and the neurites (*Figure 6A–B*), suggesting that the axonal MTs carry short glutamate side chains containing less than four glutamates. As a control, we also did staining with anti-glycylated tubulin antibodies (Gly-pep1) but did not observe any signal in embryonically derived cells, supporting the absence of polyglycylation in *C. elegans*.

Given our focus on neurite growth, we used GT335 in the following studies. As expected, TRNs extracted from *mec-12(4Es-A)* mutants showed a reduction in polyglutamylation signals and further substituting the last three glutamates in the C-terminal tail of MEC-7 with alanine (ADEDAAEAFDGE to ADADAAAAFDGA) abolished the staining signal in the axons (*Figure 6B–C*). Thus, we reasoned that both MEC-12/α-tubulin and MEC-7/β-tubulin were subjected to polyglutamylation.

Like MEC-12/α-tubulin, the MEC-7/β-tubulin C-terminal tail also contains 15 amino acids and the last 12 (430-441aa) are divergent among the isotypes (*Figure 6—figure supplement 1A*). Although changing the last three glutamates to alanine removed the polyglutamylation sites on β-tubulin, the *mec-7(3Es-A)* mutations did not induce the growth of a long ALM-PN and did not enhance the growth of ALM-PN in the *mec-12(4Es-A)* mutants (*Figure 6F*). This result suggests that polyglutamylation on β-tubulin might have weaker effects than polyglutamylation on α-tubulin.

*C. elegans* genome contains six tubulin tyrosine ligase-like genes, including *ttll-4,–5, −9,–11, –12*, and *−15*. Although *ttll-12* was thought to be the homolog of human TTLL12, which may not have glutamylase activity based on sequence divergence in the TTL domain (*van Dijk et al., 2007*), it is biochemically uncertain whether *C. elegans* TTLL-12 indeed lacks glutamate ligation activity. So, we included *ttll-12* in our analysis. Based on the single-cell RNA-sequencing data, only *ttll-4, 5*, and *12* showed significant expression in the TRNs (*Figure 6E*). To our surprise, triple knockout mutants of the three *ttll* genes only eliminated the anti-polyE signal in the TRN cell bodies but not the axons (*Figure 6B* and *Figure 6—figure supplement 1B*). Further deleting *ttll-9* and *ttll-15* to make quintuple mutants did not reduce but actually increased the staining signal in the axons. Similar anticorrelation between the polyglutamylation signals in the cell body and the axons were also seen in the *mec-12(4Es-A); mec-7(3Es-A)* mutants, in which the reduced signal in axons coincided with increased signal in cell bodies (*Figure 6C–D*). Overall, the above data appeared to suggest that the known *ttll* genes may not be responsible for the tubulin polyglutamylation in the TRN axons.

The major phenotype for the *mec-12(4Es-A)* mutant was the generation of a prominent ectopic ALM-PN, which was not observed in either *ttll-4(-)*, *5(-)*, *12(-)* triple mutants or the quintuple mutants that further deleted *ttll-9* and *15* or the sextuple mutants that deleted all six *ttll* genes (*Figure 6F*). The sextuple mutants also had anti-polyE staining of the TRN axons similar to the triple and quintuple mutants (*Figure 6—figure supplement 1C*). The phenotypic discrepancy between the tubulin mutants and the *ttll* mutants suggested the existence of other tubulin polyglutamylases and/or a function of TTLLs that is independent of tubulin glutamylation.

## MEC-12/α-tubulin detyrosination and Δ2-tubulin modification

α-tubulin detyrosination occurs at the terminal tyrosine residue, and the detyrosination can lead to the removal of the penultimate glutamate residue, generating Δ2-tubulin. α-tubulin detyrosination and its derivate Δ2-tubulin are enriched in stable MTs. Detyrosination of MTs in the axons recruits kinesin-1 motor domain and enhances kinesin-based transport (*Konishi and Setou, 2009*). Detyrosination also

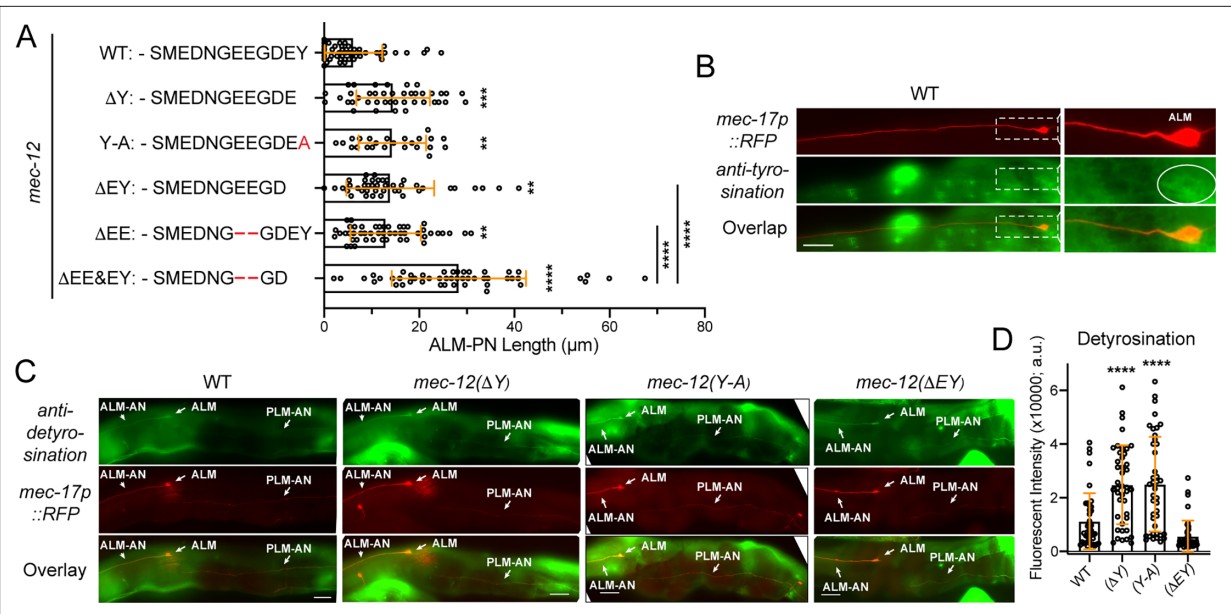

**Figure 7.** The effects of tubulin detyrosination and Δ2 modification on neurite growth. (**A**) The length of ALM-PN in strains deleting the terminal tyrosine or the last two residues. Two, three, and four asterisks correspond to p<0.05, 0.001, and 0.0001, respectively, in a post-ANOVA Tukey's HSD test. (**B**) The weak staining of wild-type TRNs with the anti-tyrosinated α-tubulin antibodies. (**C**) The staining of the wild-type animals and the various *mec-12* mutants with anti-detyrosinated α-tubulin antibodies. Arrows point to the ALM cell bodies, ALM-AN, and PLM-AN (zoom in to see the signal). (**D**) Quantification of the fluorescent intensity of the anti-detyrosination staining signal in the axons. Arbitrary units (a.u.) were used.

The online version of this article includes the following source data for figure 7:

**Source data 1.** Numeric data for *Figure 7A and D*.

protects MTs from kinesin-13-mediated depolymerization (*Peris et al., 2009*). We edited endogenous *mec-12* to first create a ΔY450 mutant that in theory would increase detyrosination levels in TRNs, although detyrosinated MTs may be retyrosinated. Second, we generated the MEC-12 Y450A mutant, which mimics permanent detyrosination. Third, we also generated the MEC-12 ΔEY mutant, which mimics permanent Δ2-tubulin.

We did not observe overt defects in TRN morphogenesis in the ΔY450, Y450A, and ΔEY mutants, although they all showed a short ectopic ALM-PN, indicating increased MT stability (*Figure 7A*). These results are consistent with the fact that detyrosination and Δ2-tubulin modification contribute to stability of MTs possibly by reducing the interaction with KLP-7/kinesin-13. Importantly, we observed combinatorial and additive effects between detyrosination and polyglutamylation, as the deletion of the terminal EY and the two Es in the GEEG motif that are subjected to polyglutamylation created a phenotype that was stronger than either ΔEY or ΔEE mutants (*Figure 7A*). The length of the ectopic ALM-PN in ΔEE&EY appeared to be the sum of the two single mutants.

Tubulin detyrosination is catalyzed by the tubulin carboxypeptidases, vasohibins (VASH1/VASH2), and their stabilizing chaperone SVBP (*Nieuwenhuis et al., 2017*). Vasohibins have no clear homologs in *C. elegans*, raising the question whether detyrosination occurs in *C. elegans*. To test this, we stained the animals with anti-tyrosinated and anti-detyrosinated α-tubulin antibodies. Using the anti-tyrosination antibodies raised against GGY, we were only able to observe weakly stained TRN cell bodies in the wild-type animals but never in theΔY450, Y450A, and ΔEY mutants (*Figure 7B*). Using the anti-detyrosination antibodies raised against GEEEGE, we could consistently observe weak staining signals in the TRNs and stronger signals in ΔY450 and Y450A, suggesting that the mutations indeed changed detyrosination levels in the neurons (*Figure 7C–D*). Interestingly, the ΔEY mutants showed weaker signal than the wild-type, likely because the terminal E of the epitope is removed from the protein, affecting the recognition by the antibodies. Our work is consistent with previous findings that mutating the terminal Y to A in TBA-1 and TBA-2 caused defects in the centration and rotation of centrosome in early embryos (*Barbosa et al., 2017*), suggesting that tyrosination/detyrosination

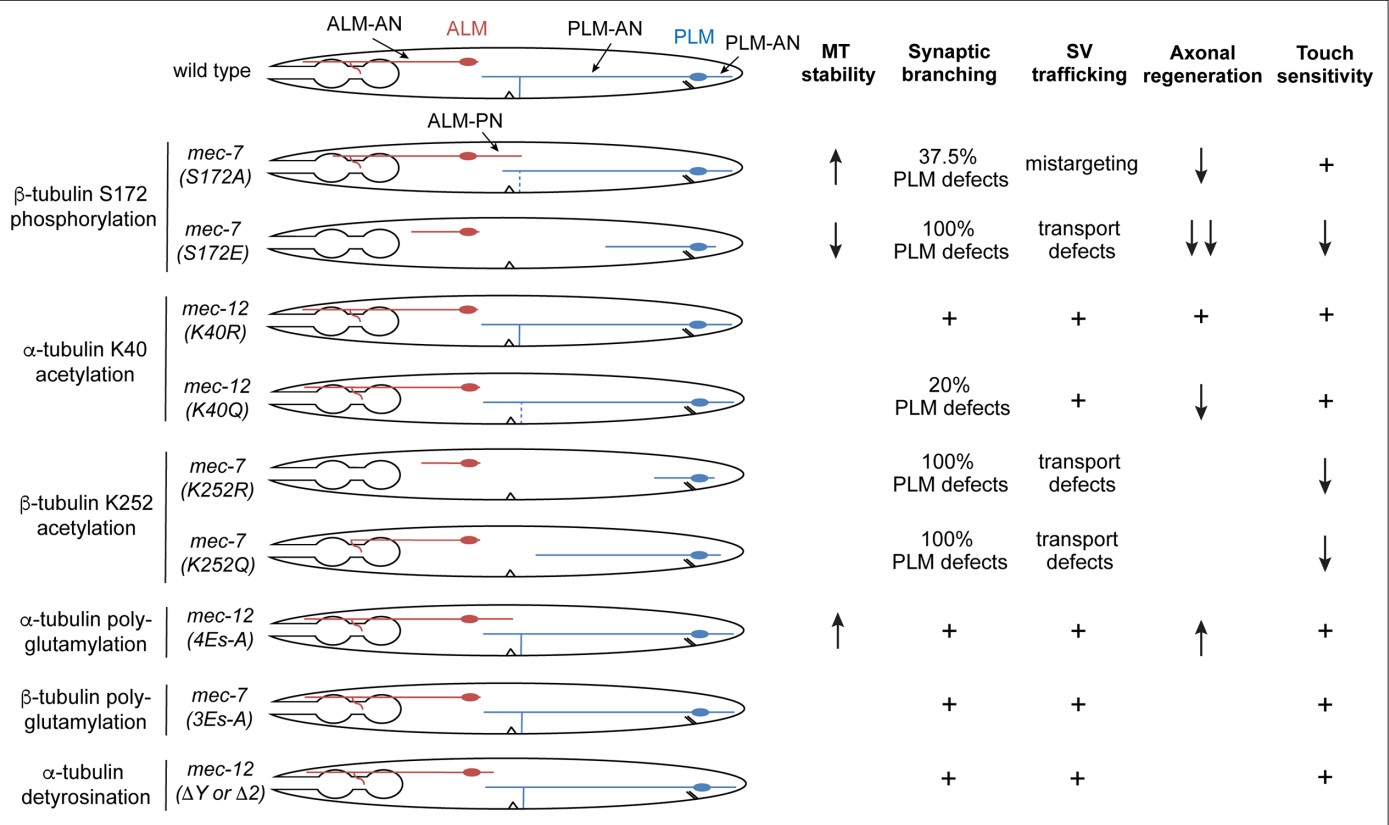

**Figure 8.** Summary of the phenotypes of the tubulin PTM site mutants. Cartoons represent the morphologies of the ALM and PLM neurons in various *mec-7* and *mec-12* mutants. Severe shortening of neurites often indicates phenotypes of antimorphic alleles, while the growth of a prominent ALM-PN is characteristic of neomorphic alleles. Dashed lines indicate partial defects of forming the PLM synaptic branch. For synaptic vesicle (SV) trafficking, mistargeting means that SV is trafficked to ALM-PN; transport defects mean that SV is mostly trapped in cell bodies or proximal segment of the neurite. MT stability is measured by EBP-2 comets; increased EBP-2 dynamics indicates reduced MT stability, and reduced EBP-2 dynamics indicates elevated MT stability. Blank space means phenotype not determined; "+" means phenotype similar to the wild-type animals.

state of tubulins may indeed regulate MT dynamics in *C. elegans*, although the enzymes that catalyze these PTMs remain elusive.

## Discussion
### Analyzing the function of tubulin PTMs in neurons with gene editing

Compared to other cell types, microtubules in neurons are highly decorated with a range of tubulin PTMs, including acetylation, polyglutamylation, detyrosination, and Δ2 modifications (*Janke and Magiera, 2020*). Whether these PTMs are simply markers for mature neurons or carry important functions during neuronal differentiation is not entirely clear. In this study, we established an in vivo model to test the functional relevance of tubulin PTM by editing the endogenous loci of neuron type-specific tubulin genes to install PTM-mimicking or nonmodifiable mutations. Leveraging the ease of genetic engineering in *C. elegans* and the simple morphology and functional readout of the mechanosensory TRNs, we investigated the role of several tubulin PTMs in axonal growth and regeneration at the single-cell level. In general, two large categories of tubulin PTMs were found (*Figure 8*). One category, such as β-tubulin S172 phosphorylation and K252 acetylation, has strong impact on neuronal morphogenesis by affecting the incorporation of tubulin heterodimers; the other category, such as tubulin polyglutamylation and detyrosintion, has subtle effects on neurite development likely by altering the interaction between MTs and MAPs. One clear theme that stemmed from these analyses is that normal neuronal development requires an optimal level of tubulin PTMs. Either the complete absence or the excessive accumulation of tubulin PTMs would cause developmental defects.

For example, eliminating S172 phosphorylation through S172A mutation led to the formation of hyperstable MTs and ectopic neurite growth, whereas mimicking permanent phosphorylation through S172E mutation resulted in highly dynamic MTs, disrupted MT polarity, and strongly impaired neurite development. Both mutations impaired axonal regeneration and caused defects in MT-dependent cargo transport. Previous studies found that S172-phosphorylated tubulin dimers cannot be incorporated into growing MTs (*Fourest-Lieuvin et al., 2006*). The fact that S172E mutants showed an antimorphic gain-of-function phenotype (indicated by severe neurite growth defects) instead of a loss-of-function phenotype (moderate growth defects found in S172P and *mec-7*(-) mutants) suggested that hyperphosphorylation did not simply inactivate the tubulin but produced dominant-negative effects. We suspect that MEC-7(S172E) proteins may form non-functional heterodimers with all α-tubulin isotypes expressed in TRNs and thus disabling all tubulins from polymerization, causing strong phenotypes. In contrast, the lack of MEC-7 in the deletion mutants may be partly compensated by other β-tubulin isotypes, resulting in a less severe phenotype.

*mec-7*/β-tubulin K252 acetylation serves as another example for the requirement of optimal PTM levels, as both the acetyl-mimicking K252Q and the unmodifiable K252R mutants caused antimorphic phenotypes. *Chu et al., 2011* previously found that acetylation of K252 slowed down tubulin incorporation into MTs, which is consistent with the K252Q phenotype. But the stronger defects in K252R mutants suggested that the complete elimination of K252 acetylation may cause more severe consequence on MT stability. Nevertheless, we could not rule out the possibility that K252R mutation induced structural changes of tubulin heterodimer independently of disabling K252 acetylation.

## α-tubulin K40 acetylation has little effects on neuronal differentiation

Although α-tubulin K40 acetylation serves as a marker for stable MTs in neurons, whether its relationship with MT stability is correlative or causative remains elusive. The loss of tubulin K40 acetyltransferase MEC-17 caused a range of defects in MT structure and organization, neurite development, and touch sensation in *C. elegans* (*Topalidou et al., 2012*). *Drosophila* α-tubulin acetylase also appeared to be required for mechanosensation in the larval peripheral nervous system (*Yan et al., 2018*). However, early studies in *Chlamydomonas* and *Tetrahymena* found that replacing wild-type α-tubulin with a nonacetylatable mutant did not cause any phenotype (*Gaertig et al., 1995*; *Kozminski et al., 1993*), and mice lacking the acetyltransferase ATAT1 or the deacetylase HDAC6 did not show obvious defects in neuronal development and functions (*Kalebic et al., 2013*; *Kim et al., 2013*). Moreover, despite in vivo evidence for the correlation of acetylation level with binding of motor protein and cargo transport in neurons (*Reed et al., 2006*), in vitro assays found that motor motility was not affected by acetylation status (*Kaul et al., 2014*). Since K40 is located in the MT lumen, its acetylation may not directly affect MAP binding.

By editing the endogenous locus of the only α-tubulin gene that codes for a K40-containing isotype, we partly solved the controversy in *C. elegans* by finding a lack of acetyltransferase mutant phenotypes in the nonacetylatable tubulin mutants. None of the morphological and functional defects in *mec-17*(-) mutants were observed in *mec-12(K40R)* mutants and only some were observed in *mec-17* enzymatically dead mutants, suggesting that MEC-17 likely functions by acetylating other substrates and/or acts independently of its enzymatic activity. The discrepancy between the phenotypes of *mec-17*(-) and *mec-12* K40 mutants highlights the need of directly assessing the effects of tubulin PTMs through endogenous gene editing.

Although our results argued against a causative relationship between K40 acetylation and MT stability during neuronal development, we could not rule out effects of K40 acetylation in other cellular conditions. For example, excessive acetylation in *mec-12(K40Q)* mutants appeared to limit the ability of the axons to branch and regenerate, indicating that acetylation may reduce polymerization dynamics when they are needed in response to external cues or stress. In addition, previous studies found that K40 acetylation increased mechanical resilience against damages caused by repetitive bending, thus protecting MTs from mechanical aging (*Portran et al., 2017*; *Xu et al., 2017*). Such function may be important in preserving axon integrity during neuronal aging. In fact, *Neumann and Hilliard, 2014* found that the loss of MEC-17 led to an age-related, adult-onset, and progressive axonal degeneration, but this phenotype appeared to be independent of MEC-17-mediated tubulin acetylation. Thus, whether K40 acetylation plays a role in neuronal aging awaits further investigation.

## Polyglutamylation and tyrosination promotes MT dynamics likely by recruiting kinesin-13

The C-terminal tail of tubulin, which protrudes from the MT surface, serves as a hotspot for PTMs that regulate the dynamic properties of MTs by fine-tuning their interaction with MAPs. For example, in vitro studies in a recombinant system found that the motility of kinesin motors were regulated by polyglutamylation, and the length of the glutamate side chain determined the type of kinesin motor that were activated (*Sirajuddin et al., 2014*). Similarly, detyrosination enhanced the interaction with kinesin-2 and kinesin-7 motor proteins, thus increasing their motility and processivity on MTs (*Barisic et al., 2015*; *Sirajuddin et al., 2014*). In our studies, we found that eliminating polyglutamylation or installing permanent detyrosination through the editing of α-tubulin locus increased MT stability and caused ectopic neurite growth. Genetic studies suggested that these effects are likely caused by reduced interaction with the MT-depolymerizing kinesin-13 motor protein, since the loss of kinesin-13 generated similar phenotypes, which were not enhanced by the tubulin PTM mutations. Moreover, our results are consistent with previous findings that kinesin-13 and polyglutamylation function in the same pathway to restrict axonal regeneration (*Ghosh-Roy et al., 2012*), while detyrosination was known to reduce the interaction with kinesin-13 (*Peris et al., 2009*). Thus, at least in *C. elegans* TRNs, both polyglutamylation and tyrosination seem to promote MT dynamics by enhancing the recruitment of kinesin-13. In fact, we observed additive effects of the two PTMs in double mutants that disabled both polyglutamylation and tyrosination.

Interestingly, removing the polyglutamylation sites in MEC-7/β-tubulin did not produce any defects and did not exacerbate the phenotypes of the *mec-12* mutants with no polyglutamylation sites, suggesting functional differences of the PTM on α- and β-tubulin. Previous studies found that α-tubulin polyglutamylation was abundant at all stages of neuronal development in mouse brain and in neuronal culture, whereas β-tubulin polyglutamylation only accumulated in mature neurons and was less abundant in general (*Audebert et al., 1994*). So, the higher abundance of α-tubulin polyglutamylation may explain the stronger effects. Alternatively, it is also possible that glutamylation on the C-terminal tail of the α-tubulin had stronger structural influences on kinesin-13 binding compared to the modification of the β-tubulin.

However, the enzymes that mediate polyglutamylation and tyrosination in the TRNs remain elusive. Deletion of the *ttll* enzymes did not eliminate the polyglutamylation signals in the TRN axons and did not cause the ectopic growth of ALM-PN as the removal of polyglutamylation sites did. Although *C. elegans* contains tubulin carboxypeptidases (CCPP-1 and CCPP-6) that catalyze deglutamylation (*Ghosh-Roy et al., 2012*; *Kimura et al., 2010*; *Klimas et al., 2023*; *O'Hagan et al., 2011*), there is no *C. elegans* homolog of the known tubulin detyrosinase. Further studies are needed to identify the potential enzymes that catalyze these tubulin PTMs.

# Materials and methods
## Strains and transgenes

*C. elegans* strains were maintained at 20 °C as previously described (*Brenner, 1974*). Strains used in this study were listed in the Key Resources Table. Some strains were provided by the *Caenorhabditis* Genetics Center (CGC) or by the National BioResource Project (NBRP) of Japan. Transgenes *uls115[mec-17p::TagRFP] IV*, *uls134[mec-17p::TagRFP] V*, *uls31[mec-17p::GFP] III*, and *zdls5[mec-4p::GFP] I* were used to visualize TRNs. Transgene *jsls609 [mec-7p::mitoGFP]* and *jsls821[mec-7p::GFP::rab-3] X* was used to visualize the transport of mitochondria and synaptic vesicles; transgene *juls338 [mec-4p::ebp-2::GFP +ttx-3p::RFP]* was used to track the dynamics of MTs. For *hpk-1* promoter-reporter, we cloned a 5 kb promoter sequence upstream of the start codon of *B* isoform of *hpk-1* and inserted it into pPD95.75 to make *hpk-1p::GFP*, which was then injected into worms to create *unkEx194 [hpk-1Bp::GFP; unc-119(+)]*. To conduct TRN-specific RNAi, we cloned the sense (without start codon) and antisense sequence of *mbk-2* and placed them downstream of a 1.9 kb TRN-specific *mec-17* promoter; both constructs were injected together to create the *unkEx101[mec-17p::mbk-2-dsRNA; ceh-22p::GFP]* transgene. OD2984 *ltSi953 [mec-18p::vhhGFP4::Zif-1] II* strain was used for TRN-specific protein degradation.

## CRISPR/Cas9-mediated gene editing

We adapted a previously published method for CRISPR/Cas9-medicated gene editing (*Dokshin et al., 2018*). Optimal single guide RNA (sgRNA) targets were found using the online tool CHOPCHOP (https://chopchop.cbu.uib.no/). The sgRNAs were synthesized using the NEB EnGen sgRNA Synthesis Kit (E3322S) and purified using NEB Monarch RNA Cleanup Kit (T2030L). One µg of the sgRNA was complexed with 20 pmol recombinant Cas9 endonuclease (NEB # M0646T) and injected into the *C. elegans* gonads with 1 µg single-stranded DNA donors as repair templates for precise editing through homologous recombination. Synonymous mutations were introduced in the repair template. Successful edits were first identified by single-worm PCR and then verified by Sanger sequencing. For each edit, two or three independent lines were obtained and examined. To create genetic knockouts, we selected Cas9 targets in the beginning exons and designed repair templates containing frameshift-causing small deletions. The CRISPR targets and repair templates used for each editing are listed in *Supplementary file 1*.

## Microscopy and statistical analysis

Unless otherwise stated, all strains subjected to data collection were bleached and grown at 20 °C for 3 days on nematode growth medium (NGM) agar plates seeded with *E. coli* OP50 bacteria. Young adults were anesthetized on 3% agarose pads containing 3% 2,3-Butanedione monoxime (BDM) and imaged using a Leica DMi8 microscope equipped with K5 sCMOS Camera. Measurements of neurite length and fluorescence intensity were made by LAS X software. To compare fluorescence intensity, strains were prepared simultaneously and imaged at the same settings (e.g. 100 ms exposure). In general, at least 40 worms were examined for each strain. Statistical analysis of one-way ANOVA followed by Dunnett's multiple comparisons test of different mutants with the wild-type animals or a Tukey's honestly significant difference (HSD) test for all pairs of conditions was performed using GraphPad Prism 8. Data were plotted as mean ± SD, and adjusted p-value <0.05, 0.01, 0.001, and 0.0001 were indicated by one, two, three, and four asterisks, respectively.

## Laser axotomy and axonal regeneration

Laser-mediated axonal cutting was adapted from a previous study (*Neumann et al., 2015*). Polystyrene microspheres (Polysciences #00876–15; 5-fold dilution) were used to immobilize late-L4 animals on 10% agarose pads. PLM axons were cut 50 µm anterior to the cell body, using a Pulsed Laser Unit attached to the Infinity Scanner on a Leica DMi8 using 63 x water lenses. To minimize injury, the lowest energy that enabled the generation of a 2- to 3 µm gap on the axon was used. Animals were then rescued and placed on NGM plates for recovery and were imaged 24 hr after the axotomy. We considered two categories of axonal regeneration: (1) reconnection, which means the proximal and distal axon segments were connected and the connection was clearly visualized in the same focal plane; (2) regrowth, which means that the proximal segment did not reconnect with the distal segment and the distal axon was often degenerated. Only the regrowth cases were used to calculate the average regrowth length, which is the length of the longest regenerative branch from the cut site. About 20 regrowth case were analyzed for each strain.

## Microtubule dynamics analysis

Day-one adult animals expressing *ebp-2::GFP* in the TRNs were immobilized as stated above and imaged with 500 ms exposure continuously for 60 s (i.e. one frame every 529ms and 115 frames in total were collected). ImageJ was used to generate kymographs by placing a 50 µm ROI on PLM-AN starting from the cell body, and the number of EBP comets in the distal 30 µm segment were counted for quantification (we found the 20 µm proximal segment near the cell body sometimes contained many very short EBP comets, obscuring the counting). To increase MT dynamics in some strains, we used a mild colchicine treatment previously reported (*Hsu et al., 2014*). L4 animals were first transferred to seeded NGM plates containing 0.125 mM colchicine and grown on these plates for 6.5 hours before being transferred back to normal NGM plates for recovery. After a one-hour recovery, the animals were mounted for video recording. In general, 15–20 animals were recorded and analyzed for each strain across multiple days.

## Recombinant proteins and kinase assays

Recombinant *C. elegans* MEC-12/MEC-7 tubulin dimers and MBK-2 kinase were produced in insect cells using the baculovirus-insect protein expression system. The *Spodoptera frugiperda* Sf9 cells

(Thermo Fisher #11496015) were used to generate and propagate recombinant baculoviruses, and the *Trichoplusia ni* ovary High Five cells (Thermo Fisher #B85502) were used for protein expression. For the tubulins, the cDNA sequences encoding MEC-12 and MEC-7 were codon-optimized for expression in High Five cells, and the recombinant tubulin heterodimer were generated using a previously published protocol (*Ti et al., 2020*).

For MBK-2, we cloned the cDNA of the *m* isoform of MBK-2 (NP_001293914.1) into the pACEBac1 vector and infected the High Five cells with P3 virus. About 60 hr after the infection, cells were harvested by centrifugation (1000 × *g*, 15 min) and lysed in lysis buffer (50 mM Tris-HCl, 20 mM imidazole, 500 mM KCl, 1 mM MgCl$_2$, 0.5 mM β-mercaptoethanol, 1 mM ATP, 1% IGEPAL, 5% glycerol, 3 U/ml Benzonase and Roche Complete EDTA-free protease inhibitor, pH 8.0) using dounce homogenization (20 strokes) on ice or at 4 °C. The extracts were clarified by centrifugation at 55,000 rpm for 1 hr. Supernatant was then filtered through the 0.22 μm Millex-GP PES membrane (Millipore) and loaded onto HisTrap HP column. After washing the HisTrap HP column with lysis buffer till the absorbance at 280 nm reaches the baseline, the bound protein was eluted with elution buffer (25 mM Tris-HCl, 250 mM imidazole, 500 mM KCl, 1 mM MgCl$_2$, 2 mM β-mercaptoethanol, 1 mM ATP, 5% glycerol, pH 8.0). The eluate was mixed with TEV protease (final 0.75 mg/ml) and dialyzed against low-salt buffer (25 mM Tris-HCl, 20 mM imidazole, 100 mM KCl, 1 mM MgCl$_2$, 2 mM β-mercaptoethanol, 1 mM ATP, 5% glycerol, pH 8.0) for 18 hr. The dialyzed protein solution was then loaded onto HisTrap HP and HiTrap Q FF columns. After washing the columns with low-salt buffer, the HisTrap HP column was disconnected, and the HiTrap Q FF column was eluted with a 0–100% gradient to the high-salt buffer (25 mM Tris-HCl, 20 mM imidazole, 500 mM KCl, 1 mM MgCl$_2$, 2 mM β-mercaptoethanol, 1 mM ATP, 5% glycerol, pH 8.0). The fractions containing MBK-2m were collected and loaded onto a Superdex 75 16/60 column (Cytiva #28989333) equilibrated in the size-exclusion buffer (1 X BRB80, 5% glycerol, 1 mM ATP, 2 mM β-mercaptoethanol, pH 6.8). The peak fractions were pooled for SDS-PAGE analysis and mass spectrometry characterization.

The kinase assay was performed as previously described (*Fourest-Lieuvin et al., 2006*; *Ori-McKenney et al., 2016*). 500 nM recombinant MEC-12/MEC-7 tubulins and a kinase candidate [500 nM recombinant MBK-2m or 475 nM CDK1/Cyclin B (Thermo Fisher; Cat# PV3292)] were incubated in MEM buffer [100 mM MES, pH 6.7, 1 mM EGTA, 1 mM MgCl$_2$, 0.1 mM DTT] supplemented with 5 mM MgCl$_2$ and 1 mM ATP or BRB80 buffer (80 mM PIPES, 1 mM MgCl$_2$, and 1 mM EGTA, pH 6.8) supplemented with 1 mM DTT, 1 mM PMSF, and 1 mM ATP at 30 °C for 1 hr. Reaction controls were set up by replacing the kinase or tubulins with water. Samples were boiled for 10 min and analyzed by western blot using the anti-phospho-S172 antibodies (abcam #ab76286; 1:3000 diluted in TBST buffer) and the anti-α-tubulin antibodies (abcam #ab7291; for loading controls) as the primary antibodies.

## Immunofluorescence

To detect tubulin PTM signals in the animals, we conducted immunofluorescent staining using a previously published protocol (*Finney and Ruvkun, 1990*). Briefly, worms fixed in 2% formaldehyde (in Ruvkun Finney Buffer) were subjected to three cycles of freeze-and-thaw. After washing twice with Tris-Triton Buffer (TTB; 100 mM Tris-HCl pH 7.4, 1% Triton X-100, 1 mM EDTA), worms were treated with 1% β-mercaptoethanol in TTB at 37 °C with gentle agitation for 4 hr. Samples were then washed with Borate buffer (25 mM H$_3$BO$_3$, 25 mM NaOH, pH 9.2) and incubated in borate buffer with 10 mM DTT at 37 °C and in borate buffer with 0.3% H$_2$O$_2$ at room temperature, sequentially; each incubation lasted 15 min under agitation and was followed by a wash using borate buffer. At this stage, worms were permeable to macromolecules due to the reduction of cuticular disulfide bonds to -SH and the subsequent oxidation of -SH to -SO$_3$. Permeabilized worms were then blocked with Antibody Buffer (PBST containing 1% BSA and 1 mM EDTA) for 1 hr and incubated with antibodies at a 1:100 (for primary antibody) or 1:1000 (for secondary antibodies) dilution in Antibody Buffer for 2 hr. Three washes with shaking for a total of 2 hr were applied after antibody incubation. The staining signal in at least 40 animals were quantified from each strain.

Antibodies used in this study included the anti-phospho-S172 (abcam; #ab76286), anti-acetyl-K40 (abcam; #ab24610), anti-polyE GT335 (AdipoGen Life Science; #AG-20B-0020-C100), anti-polyE chain IN105 (AdipoGen; #AG-25B-0030-C050), anti-glycylated tubulin Gly-pep1 (AdipoGen; #AG-25B-0034-C100), anti-tyrosinated α-tubulin (Sigma; #MAB1864-I), anti-detyrosinated α-tubulin

(Sigma; #MAB5566), and fluorophore-labeled secondary antibodies from Thermo Fisher (#A32723 and #A11006) and Jackson ImmunoResearch (#115-025-164, #111-545-003, #111-025-003, and #115-545-003).

## In vitro culture of *C. elegans* embryonic cells

*C. elegans* embryos were collected from gravid adults through a bleaching procedure, washed with M9 buffer and then egg buffer (118 mM NaCl, 48 mM KCl, 3.4 mM CaCl2, 3.4 mM MgCl2, 5 mM Hepes, pH 7.4), and incubated in egg buffer (1 ml per 100 µl egg pellet) supplemented with 0.5% chitinase (Sigma #C6137) for ~20 min until the eggshells were dissolved. Digested embryos were washed in egg buffer twice and subsequently in L-15CM medium [Leibovitz's L-15 Medium (Thermo Fisher #11415064) supplemented with 10% FBS, 100 U/ml penicillin-streptomycin (Sigma; P4333), and 0.85% sucrose]. Cells were dissociated in L-15CM using a 25-gauge needle and then adhered to slides pre-coated with 0.5 mg/ml lectin (Sigma #L0881) and 0.01 mg/ml poly-D-lysine (Thermo Fisher #A3890401). After 1 hr of adhesion, more L-15CM medium was added to the petri dish to submerge the slides and to support the growth of cells, which were examined 24 hr later.

For the subsequent immunostaining, cells were carefully washed with PBS twice, fixed with 4% formaldehyde in PBS for 8 min, and washed with ice-cold PBS. They were then permeabilized using PBST (0.1% Triton X-100 in PBS) for 10 min, blocked in the blocking buffer (4% BSA in PBST) for 30 min, and stained with antibodies (1:1000 dilution) in the blocking buffer for 2 hr; PBST wash was conducted five times after each step above. Slides were mounted using VECTASHIELD Antifade Mounting Medium (Vector Laboratories #H-1000), and coverslips were sealed by nail polish before imaging. In order to identify the TRNs among the embryonic cells, we used animals carrying a TRN marker *mec-17p::TagRFP* for the experiments and the cells that express RFP were identified as TRNs. Nevertheless, we could not distinguish the TRN subtypes in cell culture. At least 40 cultured TRNs were analyzed for each strain.

## Materials availability

All strains created in this study and newly created materials are available upon reasonable request, which can be made to the corresponding author.

## Acknowledgements

We thank the *Caenorhabditis* Genetics Center, which is funded by the National Institutes of Health (NIH) Office of Research Infrastructure Programs (P40 OD010440), and the National BioResource Project (NBRP), which is funded by the Japanese government, for providing strains. We thank Nina Peel for providing some *ttll* mutants. We thank Rui Wang in the Zheng lab for technical assistance in staining the embryonically derived TRNs. We thank Paige Wilson for technical assistance in creating the *mec-7* S172 mutants in the Martin Chalfie lab at Columbia University. This study was supported by funds from the National Natural Science Foundation of China (Excellent Young Scientists Fund for Hong Kong and Macau 32122002 to CZ), the Research Grant Council of Hong Kong [ECS 27104219, GRF 17107021, GRF 17106322, and CRF C7026-20G to CZ, and C7064-22GF to S-CT], the Food and Health Bureau of Hong Kong [HMRF 07183186 and 09201426 to CZ], and the seed fund from the University of Hong Kong [201910159087 and 202011159053 to CZ].

## Additional information

### Funding

| Funder | Grant reference number | Author |
|---|---|---|
| National Natural Science Foundation of China | Excellent Young Scientists Fund for Hong Kong and Macau 32122002 | Chaogu Zheng |
| Research Grants Council, University Grants Committee | ECS 27104219 | Chaogu Zheng |

| Funder | Grant reference number | Author |
|---|---|---|
| Research Grants Council, University Grants Committee | C7064-22GF | Shih-Chieh Ti |
| Research Grants Council, University Grants Committee | GRF 17107021 | Chaogu Zheng |
| Research Grants Council, University Grants Committee | GRF 17106322 | Chaogu Zheng |
| Research Grants Council, University Grants Committee | CRF C7026-20G | Chaogu Zheng |
| Food and Health Bureau | HMRF 07183186 | Chaogu Zheng |
| University of Hong Kong | Seed fund 201910159087 | Chaogu Zheng |
| University of Hong Kong | Seed fund 202011159053 | Chaogu Zheng |
| Food and Health Bureau | HMRF 09201426 | Chaogu Zheng |

The funders had no role in study design, data collection and interpretation, or the decision to submit the work for publication.

## Author contributions

Yu-Ming Lu, Resources, Data curation, Formal analysis, Validation, Investigation, Visualization, Methodology, Writing - original draft; Shan Yan, Resources, Data curation, Formal analysis, Validation, Investigation, Methodology; Shih-Chieh Ti, Resources, Supervision, Funding acquisition, Investigation, Methodology, Project administration; Chaogu Zheng, Conceptualization, Resources, Data curation, Formal analysis, Supervision, Funding acquisition, Investigation, Visualization, Methodology, Writing - original draft, Project administration, Writing - review and editing

## Author ORCIDs

Chaogu Zheng (ID) http://orcid.org/0000-0002-5048-4520

Reviewer #2 (Public review): https://doi.org/10.7554/eLife.94583.3.sa1
Author response https://doi.org/10.7554/eLife.94583.3.sa2

# Additional files

## Supplementary files

• Supplementary file 1. CRISPR targets and repair templates used for gene editing.
• MDAR checklist

## Data availability

All data generated or analyzed during this study are included in the manuscript and supporting files.

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

# Appendix 1

## Appendix 1—key resources table

| Reagent type (species) or resource | Designation | Source or reference | Identifiers | Additional information |
|---|---|---|---|---|
| Gene (*C. elegans*) | *mec-12* | NA | WormBase ID: WBGene00003175 | |
| Gene (*C. elegans*) | *mec-7* | NA | WormBase ID: WBGene00003171 | |
| Gene (*C. elegans*) | *klp-7* | NA | WormBase ID: WBGene00002219 | |
| Gene (*C. elegans*) | *cdk-1* | NA | WormBase ID: WBGene00000405 | |
| Gene (*C. elegans*) | *mbk-2* | NA | WormBase ID: WBGene00003150 | |
| Strain, strain background (*C. elegans* N2) | *uIs115 [Pmec-17::RFP] IV* | PMID: 26460008 | TU4065 | |
| Strain, strain background (*C. elegans* N2) | *mec-7(u1137; S172A) X; uIs115 [mec-17p::TagRFP] IV* | This study | TU6235 | Chaogu Zheng lab |
| Strain, strain background (*C. elegans* N2) | *mec-7(u1136; S172E) X; uIs115 [mec-17p::TagRFP] IV* | This study | TU6234 | Chaogu Zheng lab |
| Strain, strain background (*C. elegans* N2) | *mec-7(u1056; S172P) X; uIs115 [mec-17p::TagRFP] IV* | This study | TU6549 | Chaogu Zheng lab |
| Strain, strain background (*C. elegans* N2) | *uIs115; juIs338 [mec-4p::ebp-2::GFP +ttx-3p::RFP].* | PMID: 33378215 | CGZ562 | |
| Strain, strain background (*C. elegans* N2) | *mec-7(u1136; S172E) X; uIs115 IV; juIs338 [mec-4p::ebp-2::GFP +ttx-3p::RFP].* | This study | CGZ563 | Chaogu Zheng lab |
| Strain, strain background (*C. elegans* N2) | *mec-7(u1137; S172A) X; uIs115 IV; juIs338 [mec-4p::ebp-2::GFP +ttx-3p::RFP].* | This study | CGZ564 | Chaogu Zheng lab |
| Strain, strain background (*C. elegans* N2) | *uIs134 [mec-17p::TagRFP] V; jsIs821[mec-7p::GFP::rab-3] X* | This study | TU5595 | Chaogu Zheng lab |
| Strain, strain background (*C. elegans* N2) | *mec-7(u1136; S172E) X; jsIs821 X; uIs134 V* | This study | CGZ941 | Chaogu Zheng lab |
| Strain, strain background (*C. elegans* N2) | *mec-7(u1137; S172A) X; jsIs821 X; uIs134 V* | This study | CGZ942 | Chaogu Zheng lab |
| Strain, strain background (*C. elegans* N2) | *jsIs973 [mec-7p::mRFP +unc-119(+)] III. jsIs609 [mec7p::mtGFP +lin-15(+)] X.* | PMID: 23051668 | NM4244 | |
| Strain, strain background (*C. elegans* N2) | *mec-7(u1137; S172A) X; jsIs973 [mec-7p::mRFP +unc-119(+)] III. jsIs609 [mec7p::mtGFP +lin-15(+)] X* | This study | CGZ1019 | Chaogu Zheng lab |
| Strain, strain background (*C. elegans* N2) | *mec-7(u1136; S172E) X; jsIs973 [mec-7p::mRFP +unc-119(+)] III. jsIs609 [mec7p::mtGFP +lin-15(+)] X* | This study | CGZ1020 | Chaogu Zheng lab |
| Strain, strain background (*C. elegans* N2) | *dlk-1(ju476) I; uIs115 [mec-17p::TagRFP] IV* | This study | CGZ263 | Chaogu Zheng lab |
| Strain, strain background (*C. elegans* N2) | *uIs134; cmIs6 [(pBR104) mbk-1::GFP +pNC4.21].* | This study | CGZ70 | Chaogu Zheng lab |
| Strain, strain background (*C. elegans* N2) | *uIs134; cmEx6 [mbk-2p::GFP; rol-6(D)]* | This study | CGZ64 | Chaogu Zheng lab |
| Strain, strain background (*C. elegans* N2) | *unc-119(ed3) III; unkEx194 [hpk-1Bp::GFP; unc-119(+)]* | This study | CGZ915 | Chaogu Zheng lab |
| Strain, strain background (*C. elegans* N2) | *mbk-1(pk1389); uIs115* | This study | CGZ62 | Chaogu Zheng lab |
| Strain, strain background (*C. elegans* N2) | *mbk-2(dd5); uIs134* | This study | CGZ76 | Chaogu Zheng lab |
| Strain, strain background (*C. elegans* N2) | *mbk-2(ne992ts); uIs134* | This study | CGZ93 | Chaogu Zheng lab |

*Appendix 1 Continued on next page*

*Appendix 1 Continued*

| Reagent type (species) or resource | Designation | Source or reference | Identifiers | Additional information |
|---|---|---|---|---|
| Strain, strain background (*C. elegans N2*) | *mbk-2(ok2235) IV/nT1 [qIs51] (IV;V); zdIs5 [mec-4p::GFP] I* | This study | CGZ127 | Chaogu Zheng lab |
| Strain, strain background (*C. elegans N2*) | *mbk-1(pk1389) X; mbk-2(dd5ts) IV; uIs134 V* | This study | CGZ366 | Chaogu Zheng lab |
| Strain, strain background (*C. elegans N2*) | *mbk-1(pk1389) X; mbk-2(ne992ts) IV; uIs134 V* | This study | CGZ367 | Chaogu Zheng lab |
| Strain, strain background (*C. elegans N2*) | *mbk-1(pk1389) X; mbk-2(ok2235) IV/nT1 [qIs51] (IV;V); zdIs5 I* | This study | CGZ828 | Chaogu Zheng lab |
| Strain, strain background (*C. elegans N2*) | *mbk-1(pk1389) X; mbk-2(ok2235) IV/nT1 [qIs51] (IV;V); uIs31 III* | This study | CGZ829 | Chaogu Zheng lab |
| Strain, strain background (*C. elegans N2*) | *mbk-1(pk1389) X; uIs115 IV; unkEx101[mec-17p::mbk-2-dsRNA; ceh-22p::GFP]* | This study | CGZ404 | Chaogu Zheng lab |
| Strain, strain background (*C. elegans N2*) | *cdk-1(he5)/hT2; uIs115 IV* | This study | CGZ1178 | Chaogu Zheng lab |
| Strain, strain background (*C. elegans N2*) | *cdk-1(ok1882) III/hT2 [bli-4(e937) let-?(q782) qIs48] (I;III); uIs134 V* | This study | CGZ97 | Chaogu Zheng lab |
| Strain, strain background (*C. elegans N2*) | *cdk-1(he5)/hT2; mec-7(u1137; S172A) X; uIs115 IV* | This study | CGZ1394 | Chaogu Zheng lab |
| Strain, strain background (*C. elegans N2*) | *hpk-1(pk1393); uIs134* | This study | CGZ66 | Chaogu Zheng lab |
| Strain, strain background (*C. elegans N2*) | *mec-12(unk119; K40Q) III; uIs115 IV* | This study | CGZ814 | Chaogu Zheng lab |
| Strain, strain background (*C. elegans N2*) | *mec-12(unk120; K40R) III; uIs115 IV* | This study | CGZ815 | Chaogu Zheng lab |
| Strain, strain background (*C. elegans N2*) | *mec-17(unk126; dW) IV; uIs115 IV* | This study | CGZ1017 | Chaogu Zheng lab |
| Strain, strain background (*C. elegans N2*) | *mec-17(ok2109) IV;uIs134 V* | This study | CGZ895 | Chaogu Zheng lab |
| Strain, strain background (*C. elegans N2*) | *mec-17(ok2109) IV; atat-2(ok2415) X;uIs134 V* | This study | CGZ1098 | Chaogu Zheng lab |
| Strain, strain background (*C. elegans N2*) | *mec-17(unk126; dW) IV; atat-2(ok2415) X; uIs115 IV* | This study | CGZ1099 | Chaogu Zheng lab |
| Strain, strain background (*C. elegans N2*) | *atat-2(ok2415) X; uIs115 IV* | This study | CGZ1100 | Chaogu Zheng lab |
| Strain, strain background (*C. elegans N2*) | *mec-7(unk143; K252Q) X; uIs115 IV* | This study | CGZ1138 | Chaogu Zheng lab |
| Strain, strain background (*C. elegans N2*) | *mec-7(unk144; K252R) X; uIs115 IV* | This study | CGZ1139 | Chaogu Zheng lab |
| Strain, strain background (*C. elegans N2*) | *mec-7(unk135; 3Es-A) X; uIs115 IV* | This study | CGZ1097 | Chaogu Zheng lab |
| Strain, strain background (*C. elegans N2*) | *mec-12(unk124; Δ7aa) III; uIs115 IV* | This study | CGZ1015 | Chaogu Zheng lab |
| Strain, strain background (*C. elegans N2*) | *mec-12(unk133; 7aa-A) III; uIs115 IV* | This study | CGZ1095 | Chaogu Zheng lab |
| Strain, strain background (*C. elegans N2*) | *mec-12(unk136; 4Es-A) III; uIs115 IV* | This study | CGZ1000 | Chaogu Zheng lab |
| Strain, strain background (*C. elegans N2*) | *mec-12(unk148; 1stE-A) III; uIs115 IV* | This study | CGZ1175 | Chaogu Zheng lab |
| Strain, strain background (*C. elegans N2*) | *mec-12(unk155; 2Es-A) III; uIs115 IV* | This study | CGZ1184 | Chaogu Zheng lab |
| Strain, strain background (*C. elegans N2*) | *mec-12(unk153; 4thE-A) III; uIs115 IV* | This study | CGZ1182 | Chaogu Zheng lab |

*Appendix 1 Continued on next page*

*Appendix 1 Continued*

| Reagent type (species) or resource | Designation | Source or reference | Identifiers | Additional information |
|---|---|---|---|---|
| Strain, strain background (*C. elegans N2*) | *mec-12(unk138; 3Es-A) III; uIs115 IV* | This study | CGZ1133 | Chaogu Zheng lab |
| Strain, strain background (*C. elegans N2*) | *mec-12(unk150; Δ3E) III; uIs115 IV* | This study | CGZ1176 | Chaogu Zheng lab |
| Strain, strain background (*C. elegans N2*) | *mec-12(unk122; ΔY) III; uIs115 IV* | This study | CGZ847 | Chaogu Zheng lab |
| Strain, strain background (*C. elegans N2*) | *mec-12(unk123; Y-A) III; uIs115 IV* | This study | CGZ1014 | Chaogu Zheng lab |
| Strain, strain background (*C. elegans N2*) | *mec-12(unk121; ΔEY) III; uIs115 IV* | This study | CGZ846 | Chaogu Zheng lab |
| Strain, strain background (*C. elegans N2*) | *mec-12(unk131; ΔEE) III; uIs115 IV* | This study | CGZ1093 | Chaogu Zheng lab |
| Strain, strain background (*C. elegans N2*) | *mec-12(unk151; ΔEE&EY) III; uIs115 IV* | This study | CGZ1177 | Chaogu Zheng lab |
| Strain, strain background (*C. elegans N2*) | *mec-12(unk136; 4Es-A) III; mec-7(unk135; 3Es-A) V; uIs115 IV* | This study | CGZ896 | Chaogu Zheng lab |
| Strain, strain background (*C. elegans N2*) | *spas-1(tm683) V; uIs115 IV* | This study | CGZ1179 | Chaogu Zheng lab |
| Strain, strain background (*C. elegans N2*) | *spas-1(ok1608) V; uIs115 IV* | This study | CGZ1180 | Chaogu Zheng lab |
| Strain, strain background (*C. elegans N2*) | *klp-7(unk179; Δ11bp) III; uIs115 IV* | This study | CGZ1183 | Chaogu Zheng lab |
| Strain, strain background (*C. elegans N2*) | *klp-7(unk171; Δ14bp) III; uIs115 IV* | This study | CGZ1259 | Chaogu Zheng lab |
| Strain, strain background (*C. elegans N2*) | *klp-7(unk171; Δ14bp) III; mec-12(unk136; 4Es-A) III; uIs115 IV* | This study | CGZ1260 | Chaogu Zheng lab |
| Strain, strain background (*C. elegans N2*) | *mec-12(unk136; 4Es-A) III; uIs115 IV; juIs338* | This study | CGZ1256 | Chaogu Zheng lab |
| Strain, strain background (*C. elegans N2*) | *mec-12(unk136; 4Es-A) III; uIs115 IV; jsIs609 X* | This study | CGZ1257 | Chaogu Zheng lab |
| Strain, strain background (*C. elegans N2*) | *mec-12(unk136; 4Es-A) III; uIs115 IV; jsIs821 X* | This study | CGZ1258 | Chaogu Zheng lab |
| Strain, strain background (*C. elegans N2*) | *ttll-5(tm3360); uIs115* | This study | CGZ159 | Chaogu Zheng lab |
| Strain, strain background (*C. elegans N2*) | *ttll-5(tm3360) V; mec-12(unk136; 4Es-A) III; uIs115 IV* | This study | CGZ1393 | Chaogu Zheng lab |
| Strain, strain background (*C. elegans N2*) | *ttll-4(tm3310) III; ttll-5(tm3360) V; ttll-12(unk185; Δ5bp) II; uIs115 IV* | This study | CGZ1554 | Chaogu Zheng lab |
| Strain, strain background (*C. elegans N2*) | *ttll-4(tm3310) III; ttll-5(tm3360) ttll-9(tm3889) ttll-15 (tm3871) V; ttll-12(unk185; D5nt) II; uIs115 IV* | This study | CGZ1475 | Chaogu Zheng lab |
| Strain, strain background (*C. elegans N2*) | *ttll-4(tm3310) III; ttll-11(tm4059) IV; ttll-5(tm3360) ttll-9(tm3889) ttll-15(tm3871) V; ttll-12(unk185 d5nt) II; zdIs5 I* | This study | CGZ1474 | Chaogu Zheng lab |
| Antibody | Rabbit polyclonal anti-phospho-S172 | abcam | ab76286 | 1:100 for staining worms; 1:1000 for staining cells; 1:3000 for WB |
| Antibody | Mouse monoclonal anti-acetyl-K40 | abcam | ab24610 | Same as above |
| Antibody | Mouse monoclonal anti-polyE GT335 | AdipoGen Life Science | AG-20B-0020-C100 | Same as above |
| Antibody | Rabbit polyclonal anti-polyE chain IN105 | AdipoGen Life Science | AG-25B-0030-C050 | Same as above |
| Antibody | Rabbit polyclonal anti-glycylated tubulin Gly-pep1 | AdipoGen Life Science | AG-25B-0034-C100 | Same as above |

*Appendix 1 Continued on next page*

*Appendix 1 Continued*

| Reagent type (species) or resource | Designation | Source or reference | Identifiers | Additional information |
|---|---|---|---|---|
| Antibody | Rat monoclonal anti-tyrosinated α-tubulin | Sigma | MAB1864-I | Same as above |
| Antibody | Mouse monoclonal anti-detyrosinated α-tubulin | Sigma | MAB5566 | Same as above |
| Antibody | Goat anti-Mouse IgG, Alexa Fluor Plus 488 | Thermo Fisher | A32723 | 1:1000 for staining worms and cells |
| Antibody | Goat anti-Rat IgG, Alexa Fluor 488 | Thermo Fisher | A11006 | 1:1000 |
| Antibody | TRITC-Goat Anti-Mouse IgG | Jackson ImmunoResearch | 115-025-164 | 1:1000 |
| Antibody | Alexa Fluor 488 Goat Anti-Rabbit IgG | Jackson ImmunoResearch | 111-545-003 | 1:1000 |
| Antibody | TRITC-Goat Anti-Rabbit IgG | Jackson ImmunoResearch | 111-025-003 | 1:1000 |
| Antibody | Alexa Fluor 488 Goat Anti-Mouse IgG | Jackson ImmunoResearch | 115-545-003 | 1:1000 |
| Recombinant DNA reagent | mec-17p::mbk-2-sense | This study | CGZ#194 | Chaogu Zheng lab |
| Recombinant DNA reagent | mec-17p::mbk-2-antisense | This study | CGZ#195 | Chaogu Zheng lab |
| Recombinant DNA reagent | hpk-1bp::GFP | This study | CGZ#261 | Chaogu Zheng lab |
| Recombinant DNA reagent | hpk-1cp::GFP | This study | CGZ#262 | Chaogu Zheng lab |
| Recombinant DNA reagent | L4440-mbk-2(gDNA) RNAi | This study | CGZ#338 | Chaogu Zheng lab |
| Recombinant DNA reagent | L4440-mbk-2(cDNA) RNAi | This study | CGZ#339 | Chaogu Zheng lab |
| Recombinant DNA reagent | L4440 empty control | This study | CGZ#378 | Chaogu Zheng lab |
| Software, algorithm | LAS X Life Science Microscope Software Platform | Leica Microsystems | N/A | |

