## [Editor Report · eLife assessment]

This **fundamental** study analyzes the roles of post-translational modifications of tubulin by generating a large panel of tubulin mutants and describing their effects on morphogenesis and function of sensory neurons in *C. elegans*. The work, which is of interest to all cell biologists, in particular researchers with an interest in the microtubule cytoskeleton and neurobiology, presents conclusions that are supported by **solid** evidence. Demonstrating that all introduced mutations have the intended consequences and exploring their direct effect on microtubules would further increase the impact of the work.

---

## [Referee Report · Reviewer #2 (Public review)]

Summary:

The tubulin subunits that make up microtubules can be posttranslationally modified and these PTMs are proposed to regulate microtubule dynamics and the proteins that can interact with microtubules in many contexts. However, most studies investigating the roles of tubulin PTMs have been conducted in vitro either with purified components or in cultured cells. Lu et al. use CRISPR/Cas9 genome editing to mutate tubulin genes in *C. elegans*, testing the role of specific tubulin residues on neuronal development. This study is a real tour de force, tackling multiple proposed tubulin modifications and following the resulting phenotypes with respect to neurite outgrowth in vivo. There is a ton of data that experts in the field will likely reference for years to come as this is one of the most comprehensive in vivo analyses of tubulin PTMs in vivo.

This paper will be very important to the field, however, it would be strengthened if: (1) the authors demonstrated that the mutations they introduced had the intended consequences on microtubule PTMs, (2) the authors explored how the various tubulin mutations directly affect microtubules, and (3) the findings are made generally more accessible to non *C. elegans* neurobiologist.

(1) The authors introduce several mutations to perturb tubulin PTMs, However, it is unclear to what extent the engineered mutations affecting tubulin in the intended way. i.e. are the authors sure that the PTMs they want to perturb are actually present in *C. elegans*. Many of the antibodies used did not appear to be specific and antibody staining was not always impacted in the mutant cases as expected. For example, is there any evidence that S172 is phosphorylated in *C. elegans*, e.g. from available phosphor-proteomic data? Given the significant amount of staining left in the S172A mutant, the antibody seems non-specific in this context and therefore not a reliable readout of whether MTs are actually phosphorylated at this residue. As another example, there is no evidence presented that K252 is acetylated in *C. elegans*. At the very least, the authors should consider demonstrating the conservation of these residues and the surrounding residues with other organisms where studies have demonstrated PTMs exist.

(2) Given that the authors have the mutants in hand, it would be incredibly valuable to assess the impact of these mutations on microtubules directly in all cases. MT phenotypes are inferred from neurite outgrowth phenotypes in several cases, the authors should look directly at microtubules and/or microtubule dynamics via EBP-2 when possible OR show evidence that the only way to derive the neurite phenotypes shown is through the inferred microtubule phenotypes. For example, the effect of the acetylation or detyrosination mutants on MTs was not assessed.

(3) There is a ton of data here that will be important for experts working in this field to dig into, however, for the more general cell biologist, some of the data are quite inaccessible. More cartoons and better labeling will be helpful as will consistent comparisons to control worms in each experiment. A good example of this issue is demonstrated in Figure 2 and Figure 4:

- Fig. 2: Please label images with what is being probed in each panel

- Fig 2G is very hard to interpret-cartoon diagramming what is being observed would be helpful.

- Line 182-185: is this referring to your data or to Wu et al? It is not clear in this paragraph when the authors are describing published work versus their own data presented here.

- Fig 2!-2K is not well described. What experiment is being done here? What is dlk-1 and why did you look at this mutant?

- Figure 4C: this phenotype is hard to interpret. Where is the wt control? Where is the quantification?

- There are no WT comparison images in Figure 4I, making the quantification difficult to interpret

(4) In addition, I am left unconvinced of the negative data demonstrating that MBK does not phosphorylate tubulin. First, the data described in lines 207-211 does not appear to be presented anywhere. Second, RNAi is notoriously finicky in neurons, thus necessitating tissue specific degradation using either the ZF/ZIF-1 or AID/TIR1 systems which both work extremely well in *C. elegans*. Third, there appears to be increasing S172 phosphorylation in Figure 3 supplement 2 with added MBK-2, but there is no anti-tubulin blot to show equal loading, so this experiment is hard to interpret.

---

## [Author Response]

The following is the authors’ response to the original reviews.

**Public Reviews:**

**Reviewer #1 (Public Review):**
(1) The manuscript by Lu et al aims to study the effects of tubulin post-translational modification in *C. elegans* touch receptor neurons. Authors use gene editing to engineer various predicted PTM mutations in a-tubulin MEC-12 and b-tubulin MEC-7. Authors generate and analyze an impressive battery of mutants in predicted phosphorylation site and acetylation site of b-tubulin MEC-7, K40 acetylation site in a-tubulin MEC-12, enzymatic site of the a-tubulin acetyltransferase MEC-17, and PTM sites in the MEC-12 and MEC-7 C-tails (glutamylation, detyrosination, delta-tubulin). This represents a lot of work, and will appeal to a readership interested in *C. elegans* touch receptor neurons. The major concern/criticism of this manuscript is whether the introduced mutation(s) directly affects a specific PTM or whether the mutation affects gene expression, protein expression/stability/localization, etc. As such, this work does convincingly demonstrate, as stated in the title, that "Editing of endogenous tubulins reveals varying effects of tubulin posttranslational modifications on axonal growth and regeneration."

We thank the reviewer for the constructive comments. With regards to the major concern or criticism, we like to point out that we have previously characterized ~100 missense mutations in *mec-7* and *mec-12* (Zheng et al., 2017, PMID: 28835377; Lee et al., 2021, PMID: 33378215). So, we are familiar with the phenotypes associated with mutations that affect gene expression or protein stability, which mostly result in a null phenotype. When analyzing the PTM site mutants, we compared their phenotypes with the previously categorized phenotypes of null alleles, neomorphic mutations that increase microtubule stability, and antimorphic mutations that prevent polymerization or disrupt microtubule stability. For example, in the case of *mec-7* S172 mutations, we found that S172P mutants had the same phenotype as the *mec-7* knockout (mild neurite growth defects), suggesting that S172P likely affects protein folding or stability, resulting in the loss of MEC-7. In contrast, S172A and S172E mutations showed phenotypes similar to neomorphic alleles (the emergence of ectopic ALM posterior neurite) and antimorphic alleles (the severe shortening of all neurites in the TRNs), respectively. These phenotypic differences suggested to us that the effects of S172A and S172E mutations cannot be simply attributed to the loss of protein expression and stability. Similar logic was applied to the studies of other PTM-inactivating or -mimicking mutations.

(2) For example, the authors manipulate the C-terminal tail of MEC-12 and MEC-7, to test the idea that polyglutamylation may be an important PTM. These mutants displayed subtle phenotypes. The authors show that branch point GT335 and polyglutamyation polyE recognizing antibodies stain cultured embryonic touch receptor neurons (TRNs), but did not examine staining in *C. elegans* TRNs in situ. To my knowledge, these antibodies have not been shown to stain the TRNs in any published papers, raising the question of how these "glutamylation" mutations are affecting mec-12 and -7. The rationale for using cultured embryonic TRNs and the relevance of the data and its interpretation are not clear.

The GT335 and polyE antibodies were used by previous studies (O’Hagan et al., 2011, PMID: 21982591; and O’Hagan et al., 2017, PMID: 29129530) to detect the polyglutamylation signals in the sensory cilia of *C. elegans*. We initially tried to stain the whole animals using these antibodies but could not get clear and distinct signals in the TRNs. We reason that the tubulin polyglutamylation signals in the TRNs may be weak, and the in situ staining method which requires the antibodies to penetrate multiple layers of tissues (e.g., cuticles and epidermis) to reach the TRN axons may be not sensitive enough to detect the signal. In fact, the TRN axons are located deeper in the worm body compared to the sensory cilia that are mostly exposed to the environment. Another reason could be that the tissues (mostly epidermis) surrounding the TRN axons also have polyglutamylation staining, which makes it difficult to recognize TRN axons. This is a situation different from the anti-K40 acetylation staining, which only occurs in the TRNs because MEC-12 is the only a-tubulin isotype that carries K40. Due to these technical difficulties, we decided to use the in vitro cultured TRNs for the staining experiment, which allows both easy access of the antibodies (thus higher sensitivity) and the dissociation of the TRNs from other tissues. The fact that we were able to observe reduced staining in the *ttll* mutants and the tubulin mutants that lost the glutamate residues suggest that these antibodies indeed detected glutamylation signals in the cells.

(3) The final paragraph of the discussion is factually incorrect. The *C. elegans* homologs of the CCP carboxypeptidases are called CCPP-1 and CCPP-6. There are several publications on their functions in *C. elegans*.

We thank the reviewer for pointing out the mistake in the text. We intended to say that “there is no *C. elegans* homolog of the known tubulin carboxypeptidases that catalyze detyrosination”, which is true given that the detyrosinase vasohibins (VASH1/VASH2) homologs cannot be found in *C. elegans*. We are aware of the publications on CCPP-1 and CCPP-6; CCPP-1 is known to regulate tubulin deglutamylation in the cilia of *C. elegans* (O’Hagan et al., 2011 and 2017), while CCPP-6 may function in the PLM to regulate axonal regeneration (Ghosh-Roy et al., 2012). In the revised manuscript, we have corrected the error.

**Reviewer #2 (Public Review):**
Summary:The tubulin subunits that make up microtubules can be posttranslationally modified and these PTMs are proposed to regulate microtubule dynamics and the proteins that can interact with microtubules in many contexts. However, most studies investigating the roles of tubulin PTMs have been conducted in vitro either with purified components or in cultured cells. Lu et al. use CRISPR/Cas9 genome editing to mutate tubulin genes in *C. elegans*, testing the role of specific tubulin residues on neuronal development. This study is a real tour de force, tackling multiple proposed tubulin modifications and following the resulting phenotypes with respect to neurite outgrowth in vivo. There is a ton of data that experts in the field will likely reference for years to come as this is one of the most comprehensive in vivo analyses of tubulin PTMs in vivo.This paper will be very important to the field, however would be strengthened if: (1) the authors demonstrated that the mutations they introduced had the intended consequences on microtubule PTMs, (2) the authors explored how the various tubulin mutations directly affect microtubules, and (3) the findings are made generally more accessible to non *C. elegans* neurobiologists.(1) The authors introduce several mutations to perturb tubulin PTMs, However, it is unclear to what extent the engineered mutations affect tubulin in the intended way i.e. are the authors sure that the PTMs they want to perturb are actually present in *C. elegans*. Many of the antibodies used did not appear to be specific and antibody staining was not always impacted in the mutant cases as expected. For example, is there any evidence that S172 is phosphorylated in *C. elegans*, e.g. from available phosphor-proteomic data? Given the significant amount of staining left in the S172A mutant, the antibody seems non-specific in this context and therefore not a reliable readout of whether MTs are actually phosphorylated at this residue. As another example, there is no evidence presented that K252 is acetylated in *C. elegans*. At the very least, the authors should consider demonstrating the conservation of these residues and the surrounding residues with other organisms where studies have demonstrated PTMs exist.

We thank the reviewer for the comments. To our knowledge, there are very few phosphor-proteome data available for *C. elegans*. We searched a previously published dataset (Zielinska et al., 2009; PMID: 19530675) and did not find the S172 phosphorylation signal in MEC-7. This is not surprising, given that only six touch receptor neurons expressed MEC-7 and the abundance of MEC-7 in the whole animal lysate may be below the detection limit. However, this phosphorylation site S172 is highly conserved across species and tubulin isotypes (Figure 1-figure supplement 1 in the revised manuscript), suggesting that this site is likely phosphorylated in MEC-7.

In the case of K252, the potential acetylation site and the flanking sequences are extremely conserved across species and isotypes. In fact, the 20 amino acids from 241-260 a.a. are identical among the tubulin genes of *C. elegans,* fruit flies, Xenopus, and humans (Figure 4-figure supplement 1B). Thus, although K252 acetylation was found in the HeLa cells, this site can possibly be acetylated.

In the case of K40, we observed sequence divergence at the PTM site and adjacent sequences among the tubulin isotypes in *C. elegans*. MEC-12 is the only *C. elegans* a-tubulin isotype that has the K40 residue, and the 40-50 a.a. region of MEC-12 appears to be more conserved than other isotypes when compared to *Drosophila*, frog, and human a-tubulins (Figure 4-figure supplement 1A).

(2) Given that the authors have the mutants in hand, it would be incredibly valuable to assess the impact of these mutations on microtubules directly in all cases. MT phenotypes are inferred from neurite outgrowth phenotypes in several cases, the authors should look directly at microtubules and/or microtubule dynamics via EBP-2 when possible OR show evidence that the only way to derive the neurite phenotypes shown is through the inferred microtubule phenotypes. For example, the effect of the acetylation or detyrosination mutants on MTs was not assessed.

We thank the reviewer for the suggestions. In this study, we created >20 tubulin mutants. Due to limited time and resources, we were not able to examine microtubule dynamics in every mutant strain using EBP-2 kymographs. We assessed the effects of the tubulin mutations mostly based on the changes on neurite growth pattern. From our previous experience of analyzing ~100 *mec-7* and *mec-12* missense mutations (Zheng et al., 2017, MBoC; Lee et al., 2021, MBoC), we found that the changes in microtubule dynamics are correlated with the changes in neuronal morphologies. For example, the growth of ectopic ALM-PN is correlated with fewer EBP-2 comets and potentially reduced microtubule dynamics; this correlation holds true for several *mec-7* neomorphic missense alleles we examined before (Lee et al., 2021, MBoC) and the PTM site mutants [e.g., *mec-7(S172A)* and *mec-12(4Es-A)*] analyzed in this study. Similarly, the shortening of TRN neurites is correlated with more EBP-2 comets and increased microtubule dynamics. For the mutants that don’t show neurite growth defects, our previous experience is that they are not likely to show altered microtubule dynamics in EBP-2 tracking experiments. So, we did not analyze the acetylation mutants (which had no defects in neurite growth) and the detyrosination mutants (which had weak ALM-PN phenotype). Nevertheless, we agree with the reviewer that we could not rule out the possibility that there may be some slight changes to microtubule dynamics in these mutants.

Using tannic acid staining and electron microscopy (EM), we previously examined the microtubule structure in several tubulin missense mutants (Zheng et al., 2017, MBoC) and found that the loss-of-function and antimorphic mutations significantly reduced the number of microtubules and altered microtubule organizations by reducing protofilament numbers. These structural changes are consistent with highly unstable microtubules and defects in neurite growth. On the other hand, neomorphic mutants had only slight decrease in microtubule abundance, maintained the 15-protofilament structure, and had a more tightly packed microtubule bundles that filled up most of the space in the TRN neurite (Zheng et al., 2017, MBoC). These structural features are consistent with increased microtubule stability and ectopic neurite growth. Although we did not directly examine the microtubule abundance and structure using EM in this study, we would expect similar changes that are correlated with the neurite growth phenotypes in the PTM mutants. We agree with the reviewer, it will be informative to conduct more comprehensive analysis on these mutants using EM and other structural biology methods.

(3) There is a ton of data here that will be important for experts working in this field to dig into, however, for the more general cell biologist, some of the data are quite inaccessible. More cartoons and better labeling will be helpful as will consistent comparisons to control worms in each experiment.

Response: We thank the reviewer for the comment. In the revised manuscript, we added some cartoons to Figure 2G to show the location of the synaptic vesicles. The neurite growth phenotype should be quite straightforward. Nevertheless, we added one more Figure (Figure 8) to summarize all the results in the study with cartoons that depicted the changes to neuronal morphologies.

(4) In addition, I am left unconvinced of the negative data demonstrating that MBK does not phosphorylate tubulin. First, the data described in lines 207-211 does not appear to be presented anywhere. Second, RNAi is notoriously finicky in neurons, thus necessitating tissue-specific degradation using either the ZF/ZIF-1 or AID/TIR1 systems which both work extremely well in *C. elegans*. Third, there appears to be increasing S172 phosphorylation in Figure 3 Supplement 2 with added MBK-2, but there is no anti-tubulin blot to show equal loading, so this experiment is hard to interpret.

We added the results of *mbk-1*, *mbk-2*, and *hpk-1* mutants and cell-specific knockdown of MBK-2 into Figure 3-figure supplement 1D. Considering the reviewer’s suggestion, we attempted to use a ZIF-1 system to remove the MBK-2 proteins specifically in the TRNs using a previously published method (PMID: 28619826). We fused endogenous MBK-2 with GFP by gene editing and then expressed an anti-GFP nanobodies fused with ZIF-1 in the TRNs to induce the degradation of MBK-2::GFP. To our surprise, unlike the *mbk-2p::GFP* transcriptional reporter, the MBK-2::GFP did not show detectable expression in the TRNs, although expression can be seen in early embryos, which is consistent with the “embryonic lethal” phenotype of the *mbk-2(-)* mutants (Figure 3-figure supplement 2A-B in the revised manuscript). We reason that either endogenous MBK-2 is not expressed in the TRNs or is expressed at a very low level. We then crossed *mbk-2::GFP* with *ItSi953 [mec-18p::vhhGFP4::Zif-1]* to trigger the degradation of any potential MBK-2 proteins and did not observe the ectopic growth of ALM-PN (Figure 3- figure supplement 2C). These results suggest that MBK-2 is not likely to regulate tubulin phosphorylation in the TRNs, which is consistent with the results of other genetic mutants and the RNAi experiments.

For Figure 3 Supplement 2 (Figure 3-figuer supplement 3 in revised manuscript), because we added the same amount of purified MEC-12/MEC-7 to all reactions and had established equal loading in Figure 3E, we did not do the anti-tubulin staining in this experiment. Since higher concentration (1742 nM) of MBK-2 did not produce stronger signal than the condition with 1268 nM, we don’t think the 1268 nM band represents true phosphorylation. Moreover, the signal is not significantly stronger than the control without MBK-2 and is much lower than the signal generated by CDK1 in Figure 3E. Based on these results, we concluded that MBK-2 is not likely to phosphorylate MEC-7.

**Recommendations for the authors:**

**Reviewer #1 (Recommendations For The Authors):**
General:A summary table would help the reader digest the vast amount of phenotypic data.Cartoons to help a non-*C. elegans* reader understand the figures.

We added Figure 8 to summarize and illustrate the effects of the various mutants analyzed in this study.

Specific:The authors engineered mutations into the predicted phosphorylation site of b-tubulin mec-7. These CRISPR-alleles mutations phenocopied previously identified loss-of-function, gain-of-function, and neomorphic mec-7 alleles identified in genetic screens by the Chalfie lab. Next, the authors sought to identify the responsible kinase, taking a candidate gene approach. The most likely family - minibrain - had no effect when knocked down/out. The authors showed that cdk-1 mutants displayed ectopic ALM-PN outgrowth. Whether cdk-1 specifically acts in the TRNs was not demonstrated, calling into question whether CDK-1 phosphorylates S172 in vivo. In their introduction (lines 45-59), the authors built a case for engineering PTM mutations directly into tubulins, because the PTM enzymes may have multiple substrates. This logic applies to the cdk-1 experiment and its interpretation.

The reviewer is right. Since CDK1 and minibrain kinase are the only known kinases that catalyze S172 phosphorylation, our results suggest that CDK-1 is more likely to catalyze S172 phosphorylation in the TRNs compared to MBK-1/2. Genetic studies found that *cdk-1(-); mec-7(S172A)* double mutants did not show stronger phenotype than the two single mutants, suggesting that they function in the same pathway. Nevertheless, we could not rule out the possibility that other kinases may also control S172 phosphorylation, and the effect of CDK-1 is indirect. We mentioned this possibility in the revised manuscript.

For a-tubulin MEC-12, acetyl-mimicking K40Q and unmodifiable K40R mutants failed to stain with the anti-acetyl-a-tubulin (K40) antibody and displayed subtle TRN phenotypes. The enzymatically dead MEC-17 had phenotypes similar to those described by Topalidou (2012), confirming the Chalfie lab finding that MEC-17 has functions in addition and independent of its acetyltransferase activity. The authors moved onto a predicted acetylation site in MEC-7 and observed TRN developmental defects, and acknowledged that this may be due to tubulin instability and not a PTM. This is a concern for all mutants, as there is no way to measure whether the protein is expressed, stable, or localized properly.

We acknowledge that this is a caveat of mutational studies. An amino acid substitution at the PTM site may have multiple effects, including the change of the PTM state and potential alteration of protein conformation. Without direct evidence for enzymatic modification of the PTM site in the neurons, we could not rule out the possibility the phenotype we observed is not related to PTM and instead is the result of abnormal protein conformation and function caused by the mutation.

Nevertheless, as stated in our above response to the first point in the public review, we can phenotypically differentiate loss-of-function and gain-of-function mutants. If the mutation reduces expression or general protein stability, it is more likely to cause a loss-of-function phenotype. For most PTM site mutants, this is not the case. We observed mostly gain-of-function phenotype, suggesting that the missense mutations did not simply inactivate the tubulin protein and instead affected the functional properties of the protein.

From here, the authors manipulate the C-terminal tail of MEC-12 and MEC-7, testing the idea that polyglutamylation may be an important PTM. These mutants displayed subtle phenotypes. The authors show that branch point GT335 and polyglutamyation polyE recognizing antibodies stain cultured embryonic TRNs, but did not examine staining in TRNs. To my knowledge, these antibodies have not been shown to stain the TRNs in any published papers (see next point). The rationale for using cultured embryonic TRNs is not clear.

See our response to the second point in the public review.

Lines 548-553 There are several publications on CCPP-1 and CCPP-6 functions in TRNs and ciliated sensory neurons. SeePMID: 20519502PMID: 21982591PMID: 21943602PMID: 23000142PMID: 29129530PMID: 33064774PMID: 36285326PMID: 37287505

We thank the reviewer for pointing out these references, some of which were cited in the revised manuscript. We made a mistake in the Discussion by saying that there are no *C. elegans* homologs of tubulin carboxypeptidases while we intended to state that there is no homolog of tubulin detyrosinase in *C. elegans*. We are aware of the studies of CCPP-1 and CCPP-6 and have corrected the mistake in revised manuscript (also see our response to the third point in the public review).

**Reviewer #2 (Recommendations For The Authors):**
Figures:As stated in the public review, more cartoons and better labeling will be helpful as will consistent comparisons to control worms in each experiment. A good example of this issue is demonstrated in Figure 2 and Figure 4:(1) Figure 2: Please label images with what is being probed in each panel.

We added labels to the panels.

(2) Figure 2G is very hard to interpret - cartoon diagramming what is being observed would be helpful.

We added cartoons to help illustrate the images.

(3) Line 182-185: is this referring to your data or to Wu et al? It is not clear in this paragraph when the authors are describing published work versus their own data presented here.

It is from our data. We have made it clear in the revised manuscript.

(4) Figure 2 - 2K is not well described. What experiment is being done here? What is dlk-1 and why did you look at this mutant?

Figure 2K showed that both wild-type animals and S172A mutants could reconnect the severed axons after laser axotomy. Previous studies have found that *dlk-1(-)* mutants were not able to regenerate axons due to altered microtubule dynamics (PMID: 19737525; PMID: 23000142). We used *dlk-1(-)* mutants as a negative control, because DLK-1 promotes microtubule growth following axotomy, and the DLK-1 pathway is essential for regeneration (PMID: 23000142). We want to highlight the phenotypic difference between *dlk-1(-)* mutants and the S172E mutants. Although both mutants showed similar regrowth length, *dlk-1(-)* mutants showed unbranched regrowth probably due to the lack of microtubule polymerization, whereas the S172E mutants showed a mesh-like regrowth pattern likely due to highly dynamic and unstable microtubules. We explained the different phenotypes in the revised manuscript.

(5) Figure 4C: this phenotype is hard to interpret. Where is the wt control? Where is the quantification?

In the Figure legend, we have referred the readers to Figure 1G for the wild-type image. Quantification is provided in the text (~20% of the animals showed the branching defects).

(6) There are no WT comparison images in Figure 4I, making the quantification difficult to interpret

In the Figure legend, we have referred the readers to Figure 1A for the wild-type control. Moreover, we included a new Figure 8 to summarize the phenotypes of all mutants.

Experimental:(1) Is it clear that only MEC-7/MEC-12 are the only a- and b-tubulin present in the TRNs? The presence of other tubulins not mutated would complicate the interpretation of the results.

According to the mRNA levels, the expression of MEC-7 and MEC-12 are >100 fold higher than other tubulin isotypes. For example, single-cell transcriptomic data (Taylor et al., 2021) showed that *mec-7* mRNA is at 135,940 TPM in ALM neurons, whereas two other tubulin isotypes, *tbb-1* and *tbb-2*, have expression value of 54 and 554 TPM, respectively in the ALM. So, even if there are some other tubulin isotypes, their abundance is much lower than *mec-7* and *mec-12* and are not likely to interfere with the effects of the *mec-7* and *mec-12* mutants.

(2) The in vitro kinase assays should be quantified.

We have added the quantification.

(3) The idea that Cdk1 phosphorylates tubulin in interphase is surprising and I am left wondering how the authors propose that Cdk1 is activated in interphase. Is cyclin B (or another cyclin) present in interphase in this cell type? Expression but not activation of Cdk1 is not discussed.

CDK1 can work with cyclin A and cyclin B. *C. elegans* has one cyclin A gene (*cya-1*) and four cyclin B genes (*cyb-1*, *cyb-2.1*, *cyb-2.2*, and *cyb-3*). According to single-cell transcriptomic data of L4 animals, *cya-1* and *cyb-1* showed weak expression in many postmitotic neurons (including the ALM neurons), while *cyb-2.1*, *cyb-2.2*, and *cyb-3* had no expression in neurons. So, it is possible that *cya-1*/cyclin A and *cyb-1*/cyclin B has low level of expression in the TRNs. A previous study also found the expression of cell cycle regulators (including cyclins) in postmitotic neurons in mouse brain (Akagawa et al., 2021; PMID: 34746147).

(4) What is the significance of neurite swelling and looping in Figure 4H? The underlying cause of this phenotype is not described.

The neurite swelling and looping phenotype of *mec-17(-)* mutants were described by Topalidou *et al.*, (2012; PMID: 22658602) and were caused by the bending of the microtubules. It appears that the loss of the a-tubulin acetyltransferase altered the organization of microtubules in the TRNs. These defects were partially rescued by the enzymatically dead MEC-17, suggesting that MEC-17 may play a non-enzymatic (and likely structural) role in regulating microtubule organization. We added more explanation in the revised manuscript.

(5) It is quite surprising that polyglutamylation is not affected in the quintuple ttll mutant. Since the authors made the sextuple ttll mutant, could they demonstrate whether polyglutamylation is further reduced in this mutant via GT335 staining?

We did not make the comparison of the quintuple and sextuple *ttll* mutants because they were crossed with TRN markers with different colors for technical reasons. The quintuple mutants CGZ1475 carried *uIs115 [mec-17p::TagRFP] IV*, whereas the sextuple mutants CGZ1474 carried *zdIs5 [mec-4p::GFP] I*. As a result, we need to use different secondary antibodies for the antibody staining, which makes the results not compatible.

Polyglutmaylation signal in the cell body was strongly affected by the *ttll* mutations. In fact, in the *ttll-4(-); ttl-5(-); ttll-12(-)* triple mutants, the signal is significantly reduced in the cell body of the TRNs, as well as the cell body of other cells. What’s surprising is that the signal in the axons persisted in the *ttll* triple and quintuple mutants. As the reviewers suggested, we also stained the sextuple mutants and found similar pattern as the triple and quintuple mutants (new Figure 6-figure supplement 1C in the revised manuscript), although the results are not quantitatively comparable due to the use of secondary antibodies with different fluorophores.

Writing:(1) The beginning of the results section is quite jarring. The information in lines 96-104 should be in the Introduction.

Due to the nature of this paper, each section deals with a particular PTM. We think it is helpful to discuss some background information before describing our results on each PTM rather than giving all in the introduction. Nevertheless, we modified the beginning of the results to make it more coherent and more connected with the preceding paragraphs.

(2) Line 122-126: conclusions are not supported by the data: it is suggested from previous experiments, but authors do not look at MTs directly.

We have rephrased the statement to acknowledge that we made such conclusion based on phenotypic similarity with mutants we previously examined.

(3) I am confused by the usage of both mec-12(4EtoA) and mec-12(4Es-A). Are these the same mutations? If so, there needs to be consistency. If not, each case needs to be defined.

They are the same. We have corrected the mistake and are now using *mec-12(4Es-A)* to refer to the mutants.

Line 105: phosphor  phosphoLine 187: were  wasLine 298: is  are

The above typos are corrected.